# Intermittent bulk release of human cytomegalovirus

**Felix J. Flomm**[1,2,3,4]**, Timothy K. Soh**[1,2,3,4]**, Carola Schneider**[4]**, Linda Wedemann**[1,2,3,4]**, Hannah M. Britt**[5]**, Konstantinos Thalassinos**[5,6]**, Søren Pfitzner**[4]**, Rudolph Reimer**[4]**, Kay Grünewald**[1,3,4,7]**, Jens B. Bosse** [1,2,3,4]*

1 Centre for Structural Systems Biology, Hamburg, Germany, 2 Hannover Medical School, Institute of Virology, Hannover, Germany, 3 Cluster of Excellence RESIST (EXC 2155), Hannover Medical School, Hannover, Germany, 4 Leibniz-Institute of Virology (LIV), Hamburg, Germany, 5 Institute of Structural and Molecular Biology, Division of Biosciences, University College London, London, United Kingdom, 6 Institute of Structural and Molecular Biology, Birkbeck College, University of London, London, United Kingdom, 7 University of Hamburg, Department of Chemistry, Hamburg, Germany

* jens.bernhard.bosse@cssb-hamburg.de

**Data Availability Statement:** Raw MS data have been deposited to PRIDE with accession code PXD023444 (https://doi.org/10.6019/PXD023444). SBFSEM raw data are available on Dryad via the DOI: https://doi.org/10.5061/dryad.5dv41ns7z

## Abstract

Human Cytomegalovirus (HCMV) can infect a variety of cell types by using virions of varying glycoprotein compositions. It is still unclear how this diversity is generated, but spatio-temporally separated envelopment and egress pathways might play a role. So far, one egress pathway has been described in which HCMV particles are individually enveloped into small vesicles and are subsequently exocytosed continuously. However, some studies have also found enveloped virus particles inside multivesicular structures but could not link them to productive egress or degradation pathways. We used a novel 3D-CLEM workflow allowing us to investigate these structures in HCMV morphogenesis and egress at high spatio-temporal resolution. We found that multiple envelopment events occurred at individual vesicles leading to multiviral bodies (MViBs), which subsequently traversed the cytoplasm to release virions as intermittent bulk pulses at the plasma membrane to form extracellular virus accumulations (EVAs). Our data support the existence of a novel *bona fide* HCMV egress pathway, which opens the gate to evaluate divergent egress pathways in generating virion diversity.

## Author summary

HCMV is a clinically highly relevant virus, which causes serious disease affecting multiple organs. Despite HCMV being an important pathogen, especially for newborn children and immunocompromised patients, treatment options are still limited. Understanding how HCMV can infect a wide variety of cells is essential for developing antiviral strategies. It is well established that HCMV particles with varying glycoprotein repertoires facilitate entry into different target cells. How different glycoprotein compositions are generated at the single-particle level is still unclear. Different envelopment and egress pathways might play a role in creating this diversity.

Other data used in this publication can be accessed on Dryad through the DOI: https://doi.org/10.5061/dryad.gtht76hpt Supplementary Videos are available as supplemental information on Dryad via the DOIs: https://doi.org/10.5061/dryad.gtht76hpt and https://doi.org/10.5281/zenodo.6611135.

**Funding:** This study was funded by the Wellcome Trust (https://wellcome.org/) through a Collaborative Award (209250/Z/17/Z) to KT, KG, and JBB. KG and JBB are funded by the Deutsche Forschungsgemeinschaft (DFG, German Research Foundation, https://www.dfg.de/) under Germany's Excellence Strategy – EXC 2155 – project number 390874280. We thank the DFG for funding the lattice light sheet system through a large equipment grant to KG and JBB, project number 413831413. FJF is holding a graduate student fellowship by the Studienstiftung des deutschen Volkes (https://www.studienstiftung.de/). The Bosse lab is further supported by the DFG Research Unit FOR 5200 (BO 4158/5-1) to JBB. The Leibniz Institute of Virology is supported by the Free and Hanseatic City of Hamburg (https://www.hamburg.com/) and the Federal Ministry of Health (https://www.bundesgesundheitsministerium.de/). KG is further funded by the Free Hanseatic City of Hamburg (grant LFF-FV 71-2019). This study is part of the Leibniz ScienceCampus InterACt (Grant Agreement No. W6/2018) (https://www.leibniz-gemeinschaft.de/). The mass spectrometer used in this study was funded by a Wellcome Trust instrumentation grant 104913/Z/14/Z to KT. The funders had no role in study design, data collection and analysis, decision to publish or preparation of the manuscript. No author received a salary from any of the funders.

**Competing interests:** The authors have declared that no competing interests exist.

Here we present direct functional evidence that HCMV uses multiviral bodies (MViBs) for the bulk release of virus particles into extracellular viral accumulations (EVAs) as a novel, alternative HCMV egress pathway. Our data from two different HCMV strains, TB40/E and Merlin, indicate that MViBs play an important role in the production of viral particles. Our results provide a basis to illuminate how different egress pathways lead to varying virion compositions and potentially determine the tropism of HCMV progeny.

## Introduction

Human Cytomegalovirus (HCMV) is a ubiquitous betaherpesvirus of high clinical importance that establishes lifelong latent infection in humans. It is the leading cause of congenital disabilities in the developed world and a significant cause of disease in immunocompromised patients, such as transplant recipients, AIDS, or cancer patients (reviewed in [1]). HCMV has been ranked highest priority for vaccine development by the Institute of Medicine for over 20 years [2]. Despite continuing efforts, no approved vaccine exists so far, and antiviral therapy is currently the only treatment option, with the development of viral resistance being a significant inherent concern [3]. As HCMV causes disease affecting various tissue types and organs, understanding how HCMV can infect different cell types is essential for developing novel antiviral strategies.

It is well established that different HCMV strains have divergent surface glycoprotein compositions which correlate with their tropism and spread modes [4–8]. The trimeric complex consisting of gH/gL/gO appears to be generally needed to mediate infection by cell-free virus in all cell types and engages the cellular receptor PDGFRα on fibroblasts [9–13]. Infection of epithelial and endothelial cells by extracellular virus in addition requires the pentameric complex consisting of gH/gL/UL128/pUL130/pUL131 [4–11]. In contrast, cell-to-cell spread of HCMV seems to be independent of the trimeric complex and dependent on the pentameric complex [14]. In general, the pentameric complex guides tropism for specific cell types such as endothelial, epithelial and myeloid cells [4–8,10]. It has been shown to interact with cellular receptors such as neuropilin 2 (NRP2), the olfactory receptor family member OR14l1 and thrombomodulin (THBD) [15–17]. Extended culturing on fibroblasts such as done for the strains AD169 and Towne leads to the selection of mutations in the UL/b' region, which also codes for the pentameric complex, resulting in the loss of tropism for specific cell types in comparison to clinical strains [18–22]. For this reason, low-passage, bacterial artificial chromosome (BAC)-derived virus preparations of strains such as endotheliotropic TB40/E and pentamer-repaired Merlin [23,24] represent useful models to study HCMV egress and spread. In addition, several factors that potentially mediate the abundance of both trimeric and pentameric complexes on virions have been recently identified [8,25–31]. However, it is still unclear if these factors are sufficient to account for the observed significant variations in tropism and spread modes [24,32]. Importantly, it has been suggested that both fibroblasts and endothelial cells can release virus populations which differ in their glycoprotein content and tropism [29,33,34]. Fibroblasts seem to release both fibroblast- as well as endothelial cell-tropic virus populations while endothelial cells retain the endothelial cell-tropic population [29,34]. However, this so-called "producer-cell effect" remains controversial [32,35].

Spatio-temporally separated envelopment or egress pathways could explain how distinct virus populations are generated, but little is known about these aspects during HCMV assembly.

The virions of herpesviruses assemble in the host cytoplasm using a culminating step called secondary envelopment. During secondary envelopment, viral capsids bud into host-derived membranes, resulting in enveloped viral particles inside transport vesicles (reviewed in [36,37]. These transport vesicles subsequently release mature virions by fusing with the plasma membrane. Compared to the morphogenesis of the alphaherpesviruses Herpes simplex virus 1 (HSV-1) and Pseudorabies virus (PRV), HCMV morphogenesis is much more involved, taking not only hours but several days. During this time, the virus extensively remodels the host cell's secretory apparatus leading to the formation of the assembly compartment (AC) [38]. The AC is a dense, donut-shaped, perinuclear structure consisting of many vesicles, a well as convoluted and interconnected membranes centered around the microtubule-organizing center [39]. It contains many cellular proteins traditionally used as organelle-specific markers, including proteins originally associated with Golgi, trans-Golgi, endosomes, and lysosomes [40–43]. However, the extensive viral transformation of the cell's secretory pathways during AC formation and the short-circuiting of established cellular pathways through viral factors renders the origin of membranes and their identity less clear. Consistently, proteomics approaches indicate that the virus-induced reorganization of the secretory apparatus during AC formation leads to the mixing of membranes of different provenance as targets for secondary envelopment [44–46].

Currently, secondary envelopment events have only been shown to occur as individual events at small vesicles in the center of the AC [47,48]. This is consistent with data from alphaherpesviruses where individual virions are continuously released through the fusion of diffraction-limited, virion-containing exocytic vesicles with the plasma membrane [49,50]. In the case of HCMV, however, studies also found enveloped particles in large multivesicular structures of unknown origin. These large multivesicular structures, containing HCMV progeny, have been called endosomal cisternae or multivesicular bodies (MVBs) in the literature, even though conclusive evidence regarding their biogenesis has been lacking [40,51–53]. A recent study from the Wilson and Goodrum labs suggested that virus-containing MVBs in HCMV fibroblasts and endothelial cells are derived from membranes of different cellular origins [54].

Currently, it is unclear if virus particles in multivesicular structures represent a productive egress pathway or are instead targeted for degradation since subsequent release of virus progeny by exocytosis could not yet be demonstrated. However, it seems rather unlikely, that a cell thoroughly reprogrammed and reorganized by a virus degrades viral products in such large quantities. Interestingly, it has been shown that the deletion of the viral protein UL135 leads, among other effects, also to an abrogation of virus-filled MVB-like structures for the strain TB40/E in endothelial cells. The same mutation also led to a significant drop in viral growth, even though UL135 is dispensable for HCMV propagation in fibroblasts [18,53]. So far, no direct link between MVBs containing virus particles and virus release could be established but the current data illustrate that virus-filled MVBs occur in several HCMV-infected cell types [51–56]. In addition, mutating UL71, a viral protein likely being involved in membrane scission [57], led to an enlargement of these MVBs [52], possibly indicating their productive role. In addition, a number of other publications implicate MVBs in HCMV morphogenesis [52,54,56,58,59] and data from the related human herpesvirus 6A (HHV-6A) suggest MVB-like structures as targets for egress [60].

To provide a comprehensive view on HCMV envelopment and identify potential alternative HCMV egress routes, we here employed an integrative approach based on volumetric live-cell imaging and three-dimensional correlative light and electron microscopy (3D-CLEM) to study the egress of HCMV strain TB40 and confirmed key findings with the strain Merlin. It provided an unprecedented, spatio-temporally highly resolved view into whole HCMV infected fibroblasts. We found large transient accumulations of enveloped virions in MVB-like

structures that were positive for CD63 in line with previous reports. We dubbed these structures multi-viral bodies (MViBs) as it is unclear at this point if their biogenesis parallels *bona fide* MVBs. Importantly, we identified secondary envelopment events at MViBs, strongly suggesting that they are targets for the viral envelopment machinery. Live-cell lattice light-sheet microscopy (LLSM) showed that MViBs were transported to the plasma membrane, where they fused, and live-cell confocal microscopy illustrated that these events lead to intermittent pulses of virus bulk release. Moreover, a pH-sensitive biosensor functionally confirmed that the observed pulses were indeed due to membrane fusion events. This intermittent virus bulk release led to the formation of extracellular viral accumulations (EVAs) at the plasma membrane. Our data argue for a model in which a large fraction of HCMV capsids envelope at MViBs, which subsequently are transported to the plasma membrane where they fuse intermittently and release bulk pulses of viral particles. We propose that this pathway likely constitutes a so-far neglected HCMV egress route. Future work is needed to dissect its molecular determinants as well as its role in generating virion diversity.

## Results

### HCMV-infected fibroblasts accumulate viral material at specific extracellular sites

To get an overview of HCMV envelopment and egress routes, we initially used live-cell spinning-disk fluorescence microscopy and followed the fate of capsids and viral membranes with an HCMV-TB40 mutant expressing EGFP-labeled capsid-associated tegument protein pp150 and mCherry-labeled viral glycoprotein gM (HCMV-TB40-pp150-EGFP-gM-mCherry) [61] by single-particle tracking. Despite considerable effort and computational filtering of thousands of analyzed capsid tracks, we could not identify more than a few instances in which diffraction-limited capsid and membrane signals merged and were subsequently co-transported. While we assumed that this was due to the high signal background in the viral AC, it made us look for alternative HCMV egress and envelopment routes that we missed by focusing on trackable small individual events.

Surprisingly, we found that between 72 and 96 hpi pp150-EGFP and gM-mCherry positive virus material accumulated at defined sites in the extracellular space (Fig 1).

These sites were reminiscent of exocytosis hotspots described for HSV-1 [62] (Fig 1C) and we dubbed them extracellular viral accumulations (EVAs). At 120 hpi, 80–90% of late-infected cells had EVAs at their plasma membrane (Fig 1B).

To investigate the nature and genesis of EVAs in HCMV-TB40-infected human foreskin fibroblasts, we developed a novel three-dimensional correlative light and electron microscopy (3D-CLEM) workflow that combines spinning-disk fluorescence microscopy with serial block-face scanning electron microscopy (SBF-SEM). This approach allowed us to correlate specific labels for capsid-associated inner tegument protein (pp150) and viral membranes (gM) with volumetric EM data of whole infected cells at high resolution (Fig 1D and 1E).

We identified EVAs below infected cells by fluorescence microscopy (Fig 1C) and analyzed them by 3D-CLEM (Fig 1D and 1F and Supplementary Video 1 in https://doi.org/10.5281/zenodo.6611135). The EM workflow is based on aldehyde fixation, followed by an adapted reduced-osmium, thiocarbohydrazide, osmium tetroxide (rOTO) regimen, as well as final uranyl acetate and lead aspartate contrasting, which resulted in high contrast volumetric stacks with well-defined membrane and capsid morphology. As depicted in S1 Fig, HCMV particles in different stages of viral assembly could be clearly identified. DNA-filled virus capsids were easily recognizable as dark contoured, round to hexagonal objects, with their condensed DNA visible as a dark dot or a short line inside depending on orientation. We found that infected

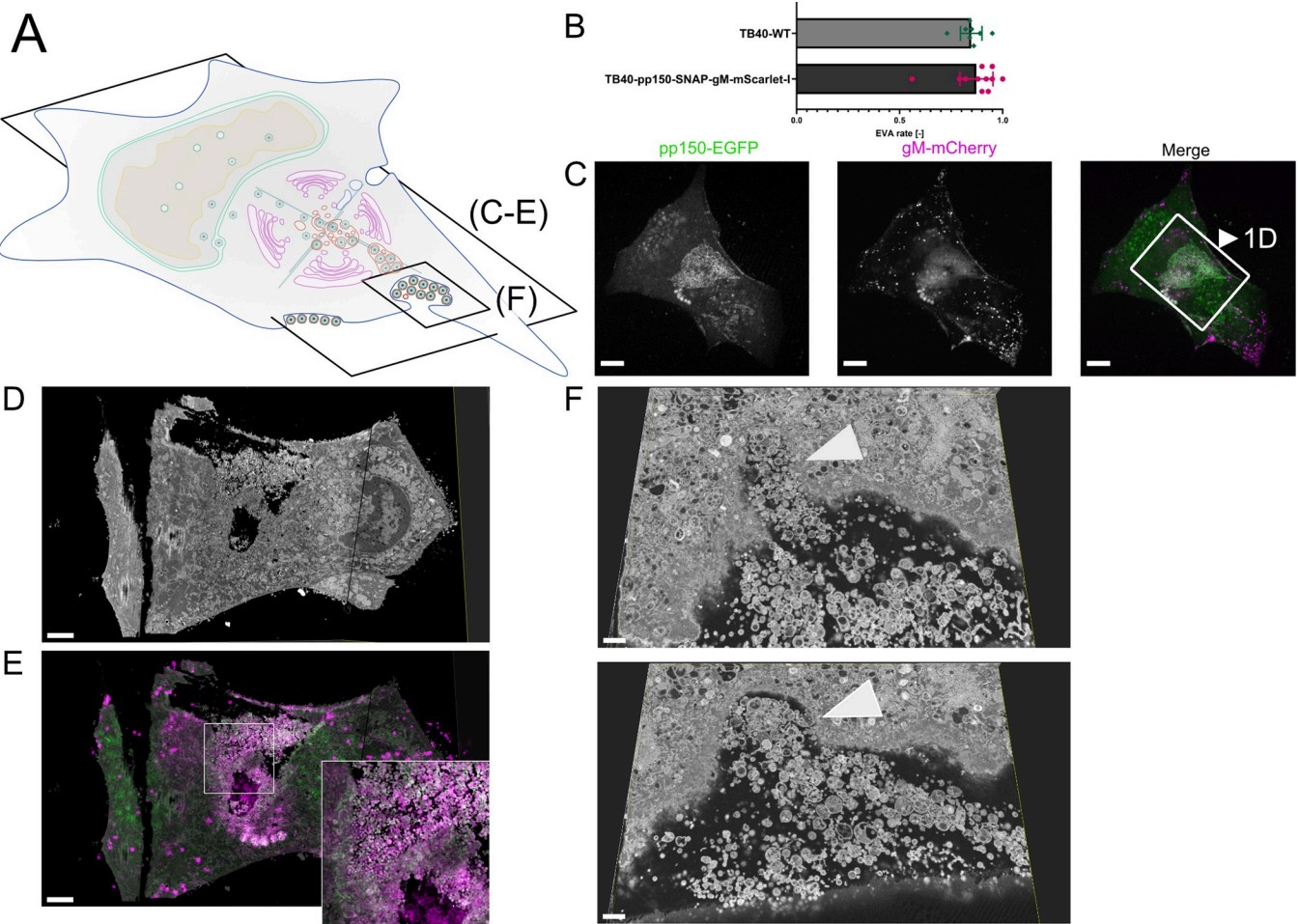

**Fig 1. EVAs are extracellular accumulations of viral products and other vesicular material. 1A** Overview indicates the subfigures' positions in relation to the whole cell. **1B** Quantification of EVA occurrence. HFF cells were infected with HCMV-pp150-SNAP-gM-mScarlet-I or HCMV-TB40-WT at an MOI of 1 and fixed at 120 hpi. HCMV-TB40-WT infected cells were stained for gB. Large overviews were acquired by spinning-disk microscopy. Late-infected cells were counted, and the rate of EVAs was quantified. Borders show the 95% confidence interval of the mean. N = 269 from 11 replicates for HCMV-pp150-SNAP-gM-mScarlet-I and N = 750 from 8 replicates for HCMV-TB40-WT. No significant difference could be found. **1C** Spinning-disk confocal section of HFF-wt cells infected with HCMV-pp150-EGFP-gM-mCherry (MOI = 3) at 4 dpi, showing EVAs positive for pp150-EGFP (green) and gM-mCherry (magenta) close to the plasma membrane. Scale bar represents 10 μm. **1D/E** CLEM of the area marked in 1C. **1D** Rendering of SBF-SEM data depicting the area close and below the plasma membrane. Scale bar represents 3 μm. **1E** Correlative overlay of SBF-SEM data from 1D with the corresponding fluorescence data from 1C indicating that pp150-SNAP and gM-mScarlet-I positive EVAs are located outside the cell. Scale bar represents 3 μm. **1F** Two z-slices are depicting invaginations (white arrowheads) next to an EVA that can be found in SBF-SEM data along the cell surface. Scale bars represent 700 nm.

cells accumulated virions, dark stained enveloped bodies, which likely represented dense bodies (capsid-free, tegument only containing particles), and a plethora of vesicles of different sizes in pp150 and gM positive EVAs (Fig 1D–1F). We also observed large invaginations at the plasma membrane that might have resulted from either endo- or exocytosis of EVAs (two slices from a volume are shown in Fig 1F).

## HCMV particles accumulate in MVBs

Next, we sought to investigate the source of EVAs. To our surprise, we found large intracellular bodies positive for both capsid and viral membrane markers at four days post-infection (dpi). A z-projection of a 3D spinning-disk microscopy stack from two adjacent cells is shown in Fig 2B, and a merge of these light and their respective EM data in Fig 2C illustrates the correlation

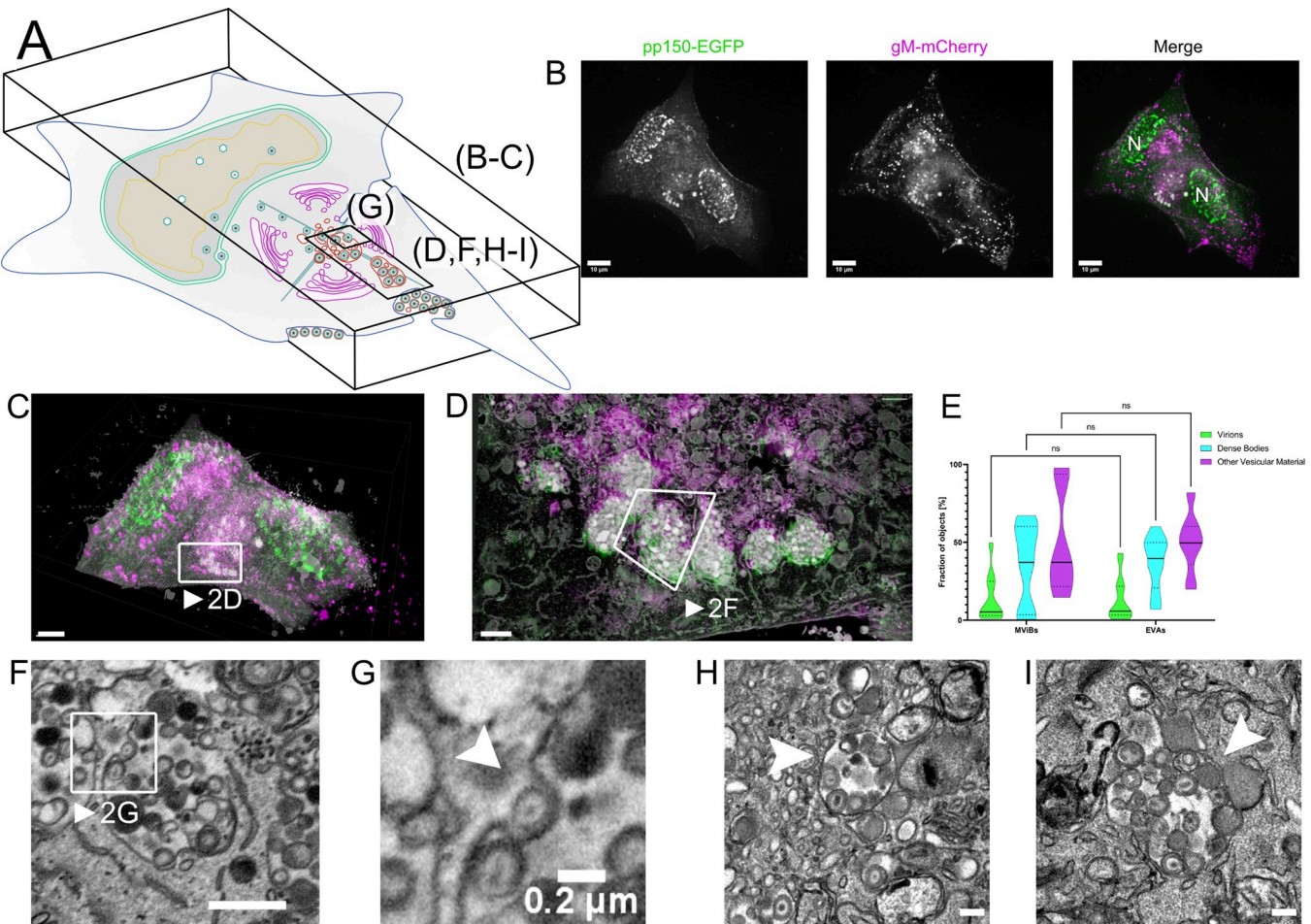

**Fig 2. Correlative fluorescence and EM detect MViBs filled with virus progeny. 2A** Overview indicates the subfigures' positions in relation to the whole cell. CLEM of HFF-cells infected with HCMV-pp150-EGFP-gM-mCherry (MOI = 3) at 4 dpi. **2B** Maximum z-projection of a 3D spinning-disk confocal microscopy stack. pp150-EGFP is colored in green and gM-mCherry signals in magenta. N marks nuclei. Scale bar indicates 10 μm. **2C** Correlative overlay of the fluorescence data shown in 2B and corresponding SBF-SEM data. Scale bar indicates 7 μm. The white frame marks MViBs. See also Supplementary Video 3B in https://doi.org/10.5281/zenodo.6611135. **2D** Correlative rendering of MViBs highlighted in 2C. The white frame marks one MViB detailed in 2F **2E** Quantitative comparison of MViB and EVA contents. Statistical analysis was performed with a 2-way ANOVA and Šídák's multiple comparisons test. Viral products and other vesicular material was found in similar ratios in both structures. No significant differences in the contents of MViBs and EVAs could be found. **2F** Section from the rendered SBF-SEM stack shown in 2D. Image signals were inverted to facilitate comparison with TEM images. An HCMV capsid potentially budding into an MViB is highlighted (white arrow). Although caution is indicated while interpreting the directionality of a process in static images from electron microscopy. Also, refer to Supplementary Videos 3A and 3B in https://doi.org/10.5281/zenodo.6611135 for a 3D rendering of the presented data. Scale bar indicates 1 μm. **2G** Insert from 2G. Arrow marks viral capsid seemingly arrested in a potential budding process. Scale bar indicates 0.2 μm. **2H-I** HFF cells were infected with HCMV-TB40-WT at an MOI of 5, fixed, and processed for EM as described for SBF-SEM at 120 hpi with the modification that the cells were embedded for classical sectioning in Epon without conductive fillers. Filled arrowheads indicate MViBs filled with virus progeny. All scale bars represent 0.2 μm.

of the datasets. Correlation of the large pp150 and gM positive spots identified in the fluorescent light microscopy data with the respective EM volumes confirmed that they represented multivesicular structures filled with significant numbers of virions, dense bodies as well as other structures (Figs 2D and 2F and 2G and S2, and Supplementary Video 2 in https://doi.org/10.5281/zenodo.6611135). To distinguish them from *bona fide* MVBs in the cell-biological sense, we dubbed them multiviral bodies (MViBs). The MViBs in our data resembled structures described in other studies [52,56], but it has been unclear if they result from a so-far unrecognized egress pathway or a degradation pathway. To elucidate if MViBs could lead to

EVAs, we quantified their contents and compared the fractions of virions, dense bodies, and other vesicular material. We found similar contents in MViBs and EVAs and no significant difference in their ratios (Fig 2E), supporting the hypothesis that EVAs are the result of MViB release. In general, MViBs were very heterogeneous in size and content. Some contained only a few particles, others up to several hundred; see also S3 Fig for an overview of a complete AC and Supplementary Videos 3A and 3B in https://doi.org/10.5281/zenodo.6611135 for representative whole-cell datasets. Supplementary Video 2 in https://doi.org/10.5281/zenodo.6611135 depicts a representative MViB that we rendered and where we segmented its contents and color-coded them as done in Fig 2E. Importantly, we found non-enveloped capsids on the surface of these MViBs (Supplementary Video 4 in https://doi.org/10.5281/zenodo.6611135). Smaller virus-containing vesicles described in previous EM-based studies [47] were often not as prominent in fluorescence microscopy but could also be found in the SBF-SEM data (S4 Fig). We also regularly found MViBs in cells infected with wild-type HCMV-TB40 (Fig 2H and 2I), confirming that MViBs are not an artifact of the fluorescently-tagged mutants. We concluded that HCMV envelopment can lead to MViBs and that EVAs had very similar contents.

## Pulses of bulk release lead to viral extracellular accumulations at the plasma membrane

To illuminate the fate of MViBs, we used two live-cell fluorescence microscopy modalities. First, we utilized inverted lattice light-sheet microscopy to acquire 3D volumes of tagged HCMV-TB40 infected cells at high temporal resolution for 15–45 minutes for minimal phototoxicity and photobleaching. We found that MViBs traveled from the assembly complex to the plasma membrane, where they seemed to fuse with the plasma membrane (Fig 3B and Supplementary Video 5 in https://doi.org/10.5281/zenodo.6611135). In a second approach, we imaged longer timespans in the infection cycle using time-lapse live-cell microscopy with less temporal coverage in 2D. To this end, we used a modified HCMV mutant with more photo-stable fluorescent tags (HCMV-TB40-pp150-SNAP-gM-mScarlet-I) for imaging z-stacks over several days. We imaged HFF cells between 72 and 96 hpi for 18 to 60 hours every 40 minutes. Strikingly, we observed MViBs coming close to the observation plane at the plasma membrane, where their appearance shifted from a round body into flattened patches of viral material (Fig 3C and Supplementary Videos 6 and 7 in https://doi.org/10.5281/zenodo.6611135). These patches were identical in their phenotype to the EVAs shown in Fig 1 and were positive for pp150 and gM. The EVAs did not diffuse away but were often left behind when cells moved away, indicating that most of the exocytosed material did not stay cell-associated. EVA formation generally occurred as intermittent pulses as MViBs came into the observation plane near the plasma membrane and fused (Supplementary Videos 6 and 7 in https://doi.org/10.5281/zenodo.6611135). The release events varied in their fluorescence intensity, consistent with our observation that MViBs were very heterogeneous in size and content. Based on these observations, we concluded that MViB exocytosis leads to EVA formation that represent the bulk exocytosis event of many virus particles at once.

## MViBs release their cargo through fusion with the plasma membrane and result in EVAs

To confirm that the observed bulk release events were indeed induced by fusion of MViBs with the plasma membrane, we used the pH-sensitive fluorescent protein super-ecliptic pHluorin as a biosensor to detect exocytosis events. We created a cell line stably expressing a CD63-pHluorin fusion construct [63] as our data (presented in the next paragraph) indicated

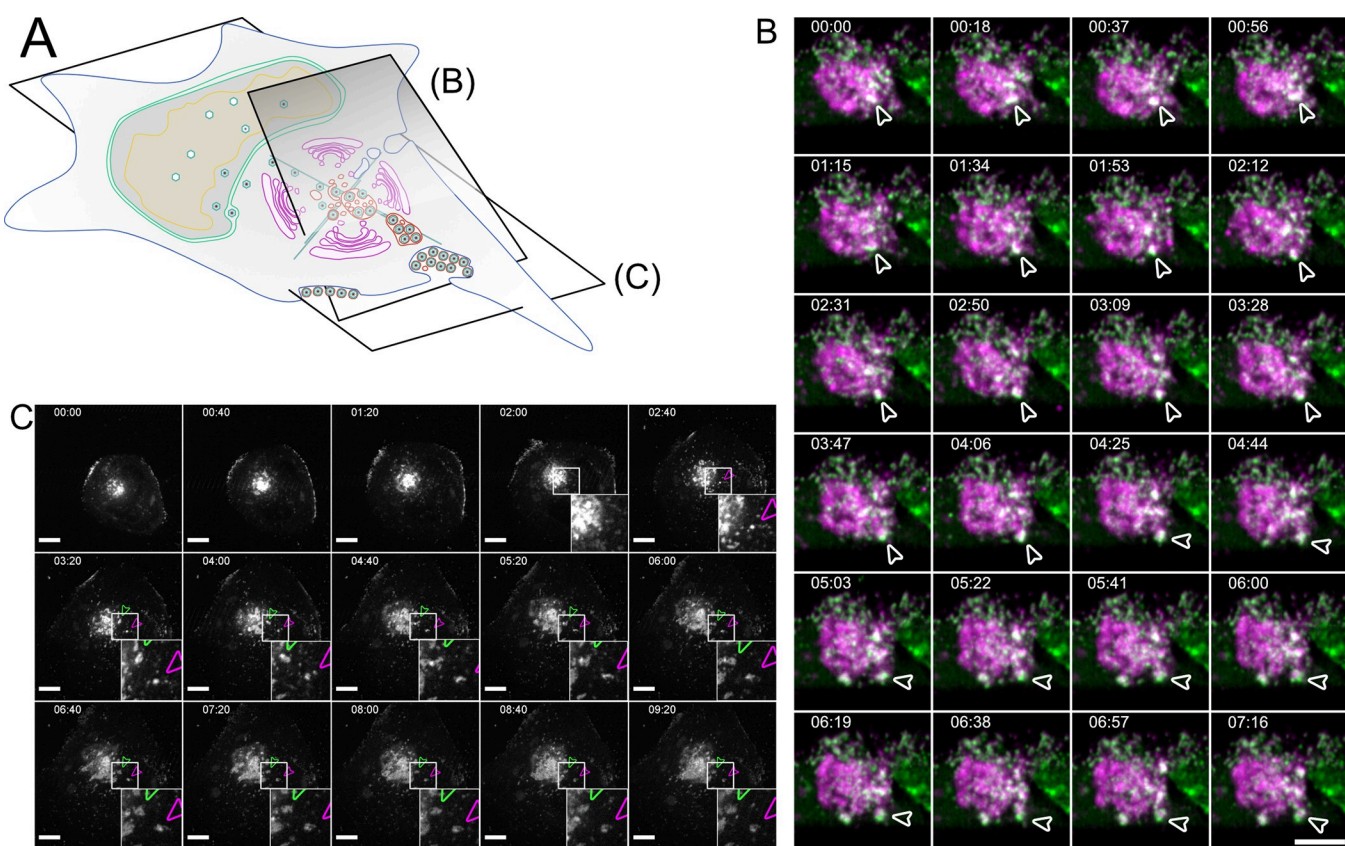

**Fig 3. Bulk release from MViBs leads to EVA formation. 3A** Overview indicates the subfigures' positions in relation to the whole cell. **3B** HFF cells were infected with HCMV-pp150-EGFP-gM-mCherry at an MOI of 1. At 96 hpi, the cells were imaged by lattice light-sheet microscopy, taking whole-cell volumes every 2.11 seconds at a 30˚ angle to the growth substrate. Maximum projections of 20 slices with a total depth of 2 μm of an area under the viral AC and incorporating the plasma membrane are shown. White arrowheads highlight an MViB positive for pp150-EGFP (green) and gM-mCherry (magenta) that approaches the plasma membrane and fuses with it. Also, refer to Supplementary Video 5 in https://doi.org/10.5281/zenodo.6611135 for a rendering and several side views. **3C** HFF cells were infected with HCMV-pp150-SNAP-gM-mScarlet-I at an MOI of 1. At 72 hpi, cells were imaged live with confocal spinning-disk microscopy. Only the gM-mScarlet-I channel is shown. Both channels can be seen in Supplementary Video 7 in https://doi.org/10.5281/zenodo.6611135. The formation of two EVAs is highlighted with white arrowheads. Scale bar indicates 10 μm. The time format is hh:mm. Also, refer to Supplementary Videos 6 and 7 in https://doi.org/10.5281/zenodo.6611135.

that CD63 is enriched on MViBs membranes but not on virions. In this construct, pHluorin is inserted into an extracellular loop of CD63, such that it points towards the luminal side in multivesicular structures and to the extracellular environment after fusion. Accordingly, pHluorin is quenched by the acidic pH inside this luminal space of MVBs, rendering the construct almost non-fluorescent. However, upon fusion with the plasma membrane, pHluorin gets exposed to the pH-neutral extracellular milieu, and fluorescence recovers rapidly. The increase in fluorescence intensity provides an easily detectable and quantifiable indicator of fusion with the plasma membrane. Moreover, we stained for gB in HCMV-TB40 infected HFF-CD63-pHluorin. In fixed, permeabilized cells the lumen of normally acidic organelles is neutralized and the intracellular pHluorin dequenched. Colocalization between CD63, gM and gB further supported the presence of CD63 on MViBs (S5A–S5C Fig and Supplementary Video 8 in https://doi.org/10.5281/zenodo.6611135).

For imaging of potential fusion events, we picked HCMV-TB40 infected cells that had not yet accumulated EVAs on the outside of the basolateral cell surface and used live-cell total internal reflection microscopy (TIRF) and lattice light-sheet microscopy to image fusion

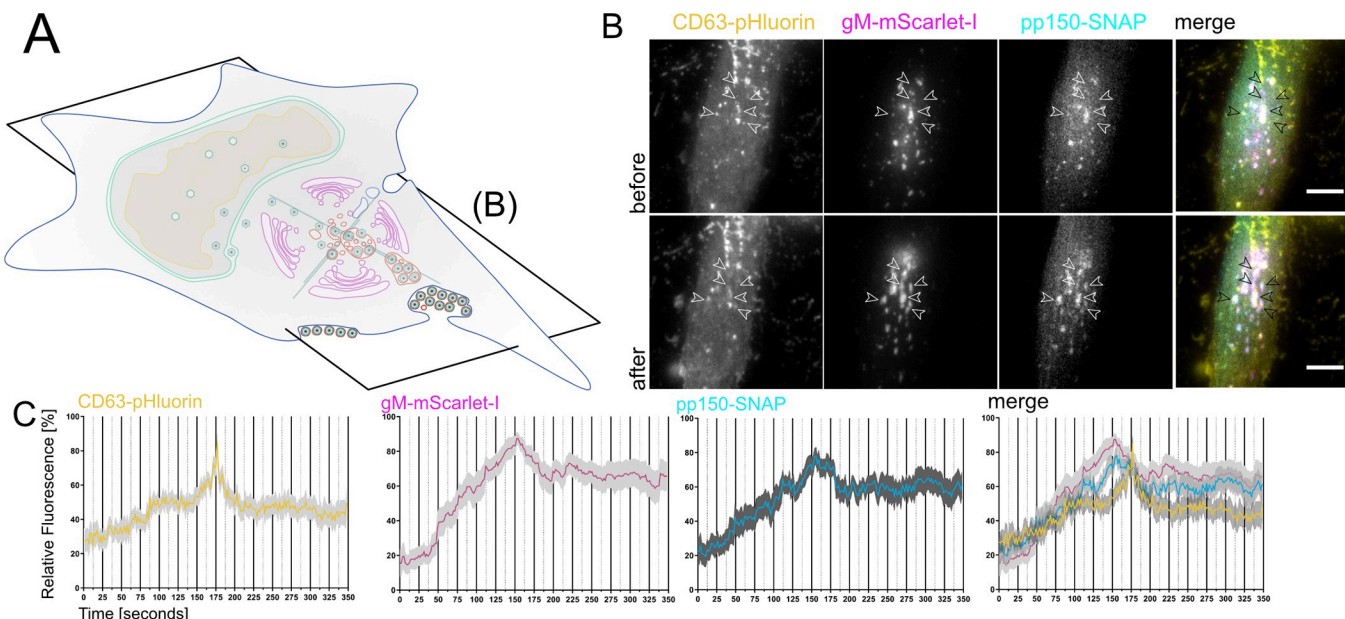

**Fig 4. EVAs are the result of fusion events between MViBs and the plasma membrane. 4A** Overview indicates the subfigures' positions in relation to the whole cell. HFF-CD63-pHluorin were infected with HCMV-pp150-SNAP-gM-mScarlet-I at an MOI of 0.6 and imaged at 72 and 96 hpi by fluorescence microscopy under TIRF conditions for 1h at an average frame rate of 0.57 frames per second (fps). **4B** TIRF images of a cell before (upper row labeled with before) and after (lower row of images labeled "after") bulk release events from MViBs occurred. Positions of EVA formation are marked by the white arrows (black in the merge). Scale bar indicates 10 μm. Also refer to Supplementary Video 9 in https://doi.org/10.5281/zenodo.6611135 for live-cell lattice light-sheet microscopy data showing the release of MViBs into EVAs. **4C** Quantification of fluorescence signals during EVA formation events over time. The fluctuation of fluorescence intensity was measured at EVA formation sites during exocytosis events. A spike of CD63-pHluorin intensity can be seen at 100 seconds (middle of the graph). Solid lines are averages from 14 EVA formation events extracted from 5 different cells chosen from 4 replicates of infections. Grey areas show the standard error of the mean.

events for several hours without phototoxicity. For TIRF microscopy, we took images every 1.5–2 seconds for 60 minutes since we predicted that actual membrane fusion and pH equilibration might be very rapid. We found that MViBs came into the TIRF-field and transitioned into EVAs shortly after arrival at the plasma membrane. MViB fusion at the plasma membrane resulted in EVAs positive for pp150 and gM (Fig 4B, arrows). Quantification of the gM-mScarlet-I and pp150-SNAP signals showed that as the vesicular bodies arrived at the plasma membrane, their fluorescence intensities increased until they peaked and subsequently fell to stable plateaus of continuously elevated signals (Fig 4C). Strikingly, these events were accompanied by flashes of green fluorescence between the MViBs arrival and EVA formation, indicating that the membranes had fused (Fig 4C). The reduction of green fluorescence indicated that most of the CD63 diffused away from the fusion site. The gM and pp150 signals increased directly before the fusion event and decreased as MViBs formed into EVAs at the plasma membrane. The exocytosed material emitted a continuously elevated signal. Using the low phototoxicity of lattice light-sheet microscopy, we also imaged volumes of cells every 15 seconds for 1h. Also here, fusion of MViBs with the plasma membrane resulted in flashes of pHluorin fluorescence, although the longer gaps between the acquisition of volumes led to displacement of the channels in fast moving objects (Supplementary Video 9 in https://doi.org/10.5281/zenodo.6611135). These results indicated that intermittent MViB fusion with the plasma membrane led to bulk release of virus material and EVA formation.

## MViBs carry markers of the endocytic trafficking system and the exosome pathway

Intermittent bulk release of vesicles is a functional hallmark of exosomal pathways. Therefore, we used immunofluorescence combined with mass spectrometry to approximate possible overlaps between virus composition and exosome generation. To this end, we performed a mass spectrometry analysis of gradient-purified extracellular virions. Gradient-purified virus particles contained markers of Golgi-to-endosome trafficking (syntaxin 12, Rab14, VAMP3), early endosomes (Rab 5C, syntaxin 7), as well as exosomes (HSP70, HSP90, GAPDH, enolase 1, 14-3-3, and PKM2) [64], suggesting that HCMV might use a mix of membranes originating from Golgi- and endosomal membranes for secondary envelopment to generate MViBs (S1 Table and S6 Fig). Our findings are consistent with a recent study, concluding that HCMV hijacks parts of the exosome pathway for egress [46].

Other classical markers for membranes used in the exosomal pathway are the tetraspanins such as CD9, CD63, and CD81. The role of CD63 in HCMV infection has been investigated before, with conflicting results [65,66]. Using immunofluorescence, we tested if the tetraspanins are localized to MViBs of HCMV-TB40 infected cells (Figs 5A–5C and S7A and S7B). The density of protein signals in the AC complicated the analysis, yet we could identify CD63 colocalizing with large vesicles containing pp150 and gM (Fig 5A–5C). We also performed EM with immunogold labeling against CD63 to investigate its presence on MViBs at high spatial resolution. Although the content of large bodies was often poorly retained after processing for immunogold staining, we regularly found HCMV particles in large bodies that also were positive for CD63, indicated by the presence of nanogold particles (S8 Fig). However, CD9 and CD81 did not specifically localize to MViBs but to the AC (S7A and S7B Fig). Besides being present on the MViB limiting membrane, exocytosed material in EVAs did not show any significant CD63 signal, implying that CD63 is unlikely to be incorporated into virions (Fig 5A–5C). This observation is supported by the absence of CD63 in our virion proteomics data (S1 Table) and is in line with previous studies [67].

To gain further insight into the biological identity of MViBs, we tested if HCMV-TB40 bulk release was susceptible to inhibitors of MVB biogenesis or exosome release. Out of an initial panel of ten drugs (Bexin-1, Simvastatin, Climbazole, GW4869, Ketotifen, Manumycin A, Nexinhib20, Suphisoxazole, Tipifarnib, U18666A) that were described to influence MVB or exosome biogenesis, we characterized the effect on HCMV for three of them (Ketotifen, Tipifarnib, U18666A) in more detail (S9 Fig). U18666A is an inhibitor of cholesterol trafficking [68,69] Tipifarnib is a farnesyl transferase inhibitor with high activity against exosome production [70] and Ketotifen is a mast-cell stabilizing agent, currently under investigation for its ability to block exosome release from cancer cells [71,72]. We tested these drugs at concentration ranges between 0.1x and 2x of reported active concentrations from the literature during virus infection. Only Tipifarnib was able to significantly reduce viral titers at 4 dpi (S9C Fig). Tipifarnib was also able to reduce the number of EVAs present at 5 dpi (S9A and S9B and S9D Fig). Moreover, we found no significant cytotoxicity of Tipifarnib in our HFF cells compared to the vehicle control (S9E Fig). While Tipifarnib had no pronounced effect on the expression of the immediate-early genes IE1/2 or the early gene UL44, it had a significant effect on the abundance of the late protein pp150 which is a inner tegument protein (S9F Fig) in total cell lysates. Using Ketotifen, Tipifarnib, and U18666A, we were, therefore, unable to delineate the biogenesis of MViBs in more detail.

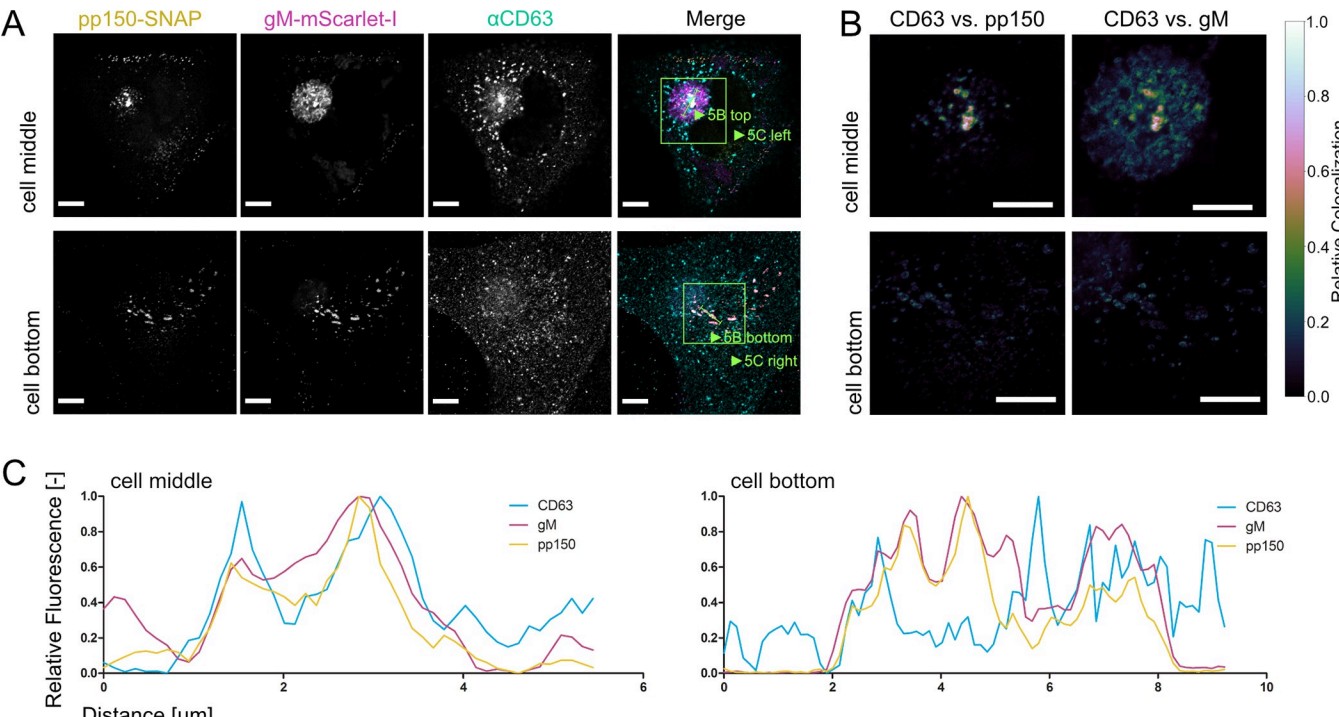

**Fig 5. Tetraspanin CD63 localizes to MViBs. 5A** HFF cells were infected at an MOI of 1 with HCMV-pp150-SNAP-gM-mScarlet-I, fixed at 4 dpi, stained for CD63, and whole cells were imaged using confocal laser scanning microscopy. From a representative cell, two slices are shown. One slice depicts the middle of the cell (cell middle), and one depicts the plasma membrane level (cell bottom). The fluorescence pattern of CD63 (αCD63) was compared to gM (gM-mScarlet-I), and pp150 (pp150-SNAP). In the cell's center, CD63 localized to the assembly complex' center and marked MViBs in the cytoplasm, which were pp150 and gM positive. At the plasma membrane, EVAs were positive for pp150 and gM signals but lacked CD63. **5B** Spatially weighted colocalization analysis shows areas in the assembly complex where CD63 colocalization with MViBs is especially pronounced (cell middle). No significant colocalization between CD63 and pp150 or gM is present in EVAs (cell bottom). All scale bars (5A and 5B) indicate 10 μm. **5C** Line plots for the indicated areas in 5B.

## A repaired BAC-derived HCMV-Merlin produces MViBs but exhibits reduced EVA formation

To validate that bulk release is not a sole phenotype of HCMV-TB40, we tested if MViBs and EVAs can also be found in cells infected with BAC-derived HCMV-Merlin which expresses both RL13 and UL128 [24,73,74]. We detected EVAs on HCMV-Merlin-infected HFFs, although at lower frequency (12–15%) compared to HCMV-TB40 (80–90%) (Figs 6B vs. 1B). Fig 6C depicts a representative example of an EVA-containing, HCMV-Merlin-infected HFF cell. Importantly, we frequently found large intracellular bodies positive for pp150 and gM (Figs 6D and S10) in HCMV-Merlin-infected HFFs. CLEM confirmed that they represented MViBs (Figs 6D and 6E and S11). These data suggest that HCMV-Merlin can also release viral progeny by bulk release (Figs 6D and 6E and S11).

In summary, our data indicate a novel functional egress pathway for HCMV in which MViBs are first generated by envelopment of several viral capsids at the same vesicle, and subsequently intermittently exocytosed at the plasma membrane, leading to the bulk release of a large number of viral particles into EVAs. However, it is still unclear which host membranes are targeted to generate MViBs, although it is likely that membranes from multiple organelles, including endocytic membranes are mixed together for viral envelopment [41,43–45]. Future studies are needed, to illuminate the molecular determinants and a potential role of MViBs in releasing virions of specific cell-tropism.

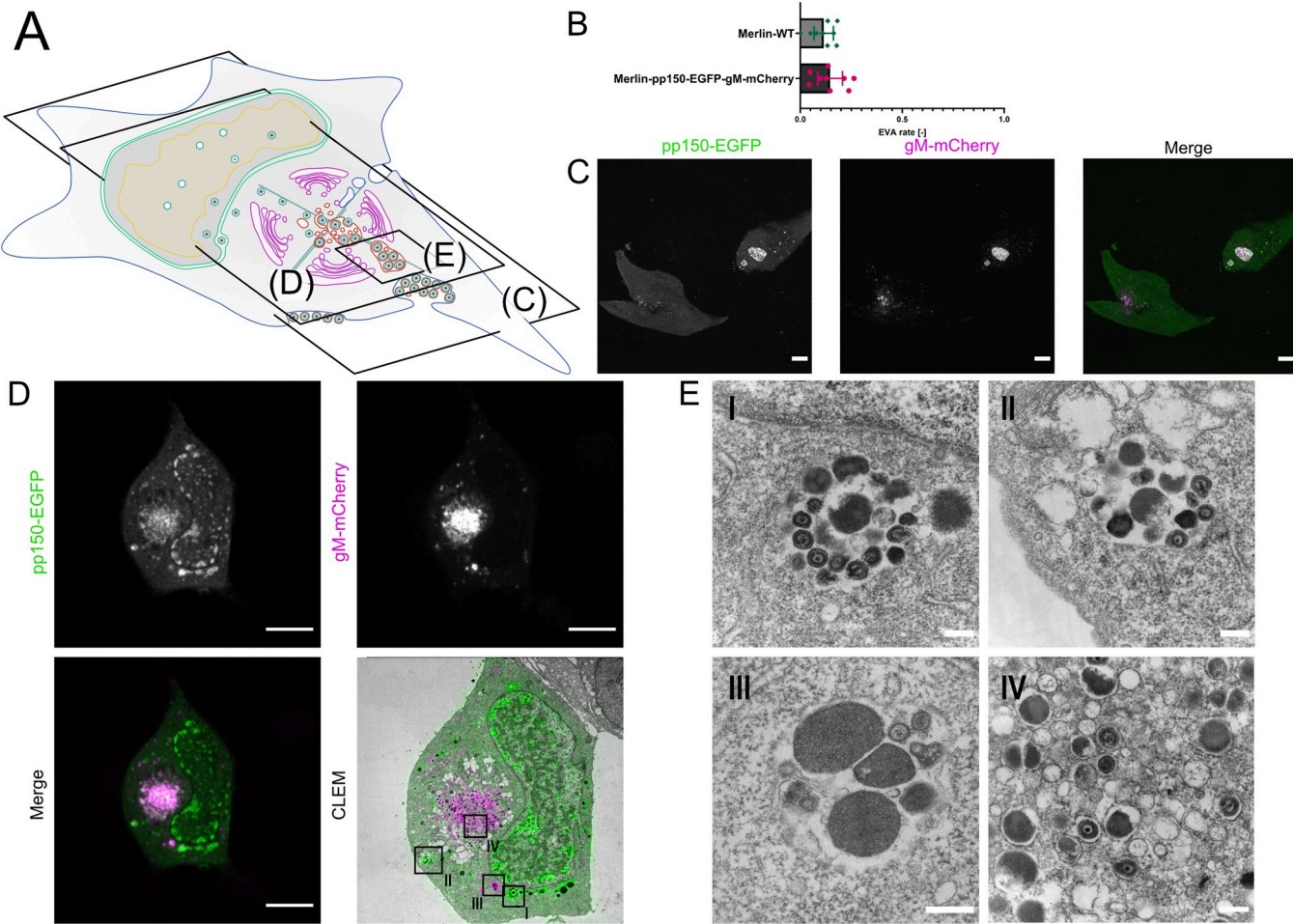

**Fig 6. HCMV Merlin produces MViBs and EVAs. 6A** Overview indicates the subfigures' positions in relation to the whole cell. **5B** HFF cells were infected with HCMV-Merlin-pAL1502-WT or HCMV-Merlin-pAL1502-pp150-EGFP-gM-mCherry at an MOI of 1. The cells were fixed at 120 hpi and the samples infected with the untagged virus were IF-stained against gB. Data acquisition and EVAs quantification was performed as described for HCMV-TB40. Shown is the rate of EVAs in late infected cells. Error bars indicate the 95% confidence interval of the mean from 9 replicates each. Values for N (counted late infected cells) are as follows: Merlin-WT: 327, Merlin-pp150-EGFP-gM-mCherry: 333. **6C** Single z-section from the dataset acquired for the analysis in Fig 6B. Shown are the lower plasma membranes of HCMV-Merlin-pAL1502-pp150-EGFP-gM-mCherry infected cells. Bright patches positive for pp150 (green) and gM (magenta) indicate the presence of EVAs. **6D** CLEM of an HFF cell infected with HCMV-Merlin-pAL1502-pp150-EGFP-gM-mCherry, infected at an MOI of 1, fixed and imaged at 4 dpi. Large cytoplasmic bodies positive for pp150 and gM were investigated with EM (black frames in the CLEM panel; see Fig 6E) Scale bars indicate 10 μm. **6E** High magnification TEM images from the frames shown in the CLEM panel of Fig 6D. Shown in I-III are large MViBs, filled with virus products similar to the MViBs found in TB40 (Fig 2). Panel IV shows a section of the AC with single-enveloped viral products. Scale bars indicate 250 nm.

## Discussion

Little data exist on the spatio-temporal organization of HCMV egress at the subcellular level. Previous studies have mostly reported single-virion/single-vesicle envelopment events, which have shaped our current picture of HCMV secondary envelopment [47,52]. These data are consistent with a study by Hogue et al. [75], which shows that individual alphaherpesvirus virions are released at the plasma membrane. Still, data suggests that virus-filled multivesicular structures can form in HCMV-infected cells as well as HHV-6A and Murine Cytomegalovirus (MCMV) infected cells [40,52,55,56,60,76] and that Golgi- and endosome-derived membranes are targeted by HCMV [40–43,51,67]. While the previous literature often called these virus-containing multivesicular structures "MVBs", we decided to dub them multi-viral bodies (MViB) as we could not untangle their descent clearly. A recent study from the Wilson and

Goodrum labs suggested that virus-containing structures in HCMV fibroblasts and endothelial cells are derived from membranes of different cellular origins [54]. Importantly, a functional role for these "virus-containing MVBs" or "MViBs" in egress has lacked so far [40,51–53].

Here, we started by investigating EVAs. We were intrigued that most cells infected with HCMV-TB40 were positive for EVAs late in infection and investigated their formation. Using a novel 3D-CLEM workflow that correlates dynamic information from spinning-disk fluorescence microscopy with high-resolution information from serial block-face scanning electron microscopy, we found that HCMV-TB40 likely produces virus particles by budding into MViBs. By time-lapse and functional live cell imaging, we provide evidence that MViBs can fuse with the plasma membrane and intermittently release tens to hundreds of virus particles in bulk, resulting in plasma membrane-associated EVAs. Proteomics of purified virions, functional imaging, and correlation of CD63 localization with MViBs suggested that MViB-mediated HCMV-TB40 egress might use features of the cellular exosomal pathway; however, drugs inhibiting MVB formation and exosome release showed no or inconclusive effects. Finally we explored the occurance of MViBs in HCMV-Merlin-infected HFF cells and could confirm that they also harbored MViBs, although release into EVAs seemed to be reduced.

While EVAs represented static endpoints, MViBs in HCMV-TB40 infected HFFs were highly dynamic and transient. In contrast, we observed less EVAs but frequent appearance of MViBs in HCMV-Merlin-infected cells, suggesting a prolonged retention of MViBs in cells infected with this strain. Integrating imaging technologies that can cover large spatio-temporal ranges of HCMV infection proved to be instrumental in analyzing the role of MViBs in HCMV egress. Because electron microscopy is generally incompatible with living cells, no dynamic information can be obtained. Correlative light microscopy can ideally complement high-resolution EM imaging by supplementing specific labels and temporal context. Since it is a correlative method, results are ultimately inferred and need careful evaluation. Our live-cell imaging indicated that MViBs in HCMV-TB40 form relatively quickly between 72 and 96 hpi and were rapidly released asynchronously, leading to pulses of EVA formation. This mechanism is in contrast to studies that have been performed in alphaherpesviruses, where single PRV virus particles have been shown to travel to the plasma membrane and be released by fusion [49]. Our data, however, does not exclude the existence of a separate egress pathway, in analogy to the mechanisms shown for alphaherpesviruses (reviewed in [77]). Compared to previous studies, our new correlative 3D-CLEM workflow provides a major technological advancement permitting us to observe whole cells in a defined infection state without the need for serial sectioning [47]. This has allowed us to analyze transient MViBs in infection with HCMV-TB40, which would have been otherwise hard to catch at high resolution. However, the frequent appearance of MViB-like structures in light microscopy images of HCMV-Merlin-infected cells allowed for easy characterization of these structures in CLEM. The aforementioned reduced EVA prevalence potentially indicates that MViB-mediated release of HCMV-Merlin progeny is attenuated in our fibroblast model. This might explain why HCMV-Merlin spread through the cell-free route is strongly diminished for HCMV-Merlin variants expressing UL128 and RL13 such as the BAC-derived pAL1502 HCMV-Merlin used here [24,73,74]. Still, future studies are needed to explore the link between strain-dependent differences in bulk release and spreading behavior [24,32,73,78,79].

Moreover, it remains unclear from our data if HCMV uses *bona fide* cellular MVBs for envelopment and transforms them into MViBs or if they are generated *de novo*. Since HCMV intensively remodels the cells secretory apparatus, it seems unlikely that traditional organelle identity and function is applicable in the infected cell [41,44,45,80]. Therefore, one might even argue that the distinction between the use of *bona fide* MVBs and virus-induced *de novo* generation is unnecessary, or even impossible. Nevertheless, cellular MVBs can produce similar

bulk pulses of extracellular vesicles (EVs) or exosomes by fusion with the plasma membrane, which makes factors involved in their function potentially interesting in the context of our findings [81–83]. EVs form through budding into the lumen of late endosomes. This process generates MVBs characterized by the presence of the late endosomal markers CD63, LAMP1, LAMP2, Rab4, and Rab5 (reviewed in [83]). Budding at MVBs is catalyzed by the endosomal sorting complex required for transport (ESCRT) [83]. While some parts of the ESCRT machinery play a role in the secondary envelopment of alphaherpesviruses [57,84–86], they might not play a role in HCMV infection [87–89]. However, it was recently shown that HSV-1 proteins pUL7 and pUL51 form a complex that might constitute a mimic of an ESCRT-III complex. HCMV homologs pUL103 and pUL71 are predicted to be structurally very similar to their HSV-1 counterparts and might likewise perform ESCRT functions for the virus during infection [57]. A recent proteomics study supports this notion by showing that HCMV utilizes parts of the exosome biogenesis machinery independently of classical ESCRT-pathways [46].

Members of the tetraspanin family, such as CD9, CD81, and CD63, have also been described to be enriched on EV membranes [81]. Tetraspanins are known to form microdomains called tetraspanin-enriched microdomains on the cell surface [90] and are active in the organization of the plasma membrane, recycling, and cellular signaling [90,91]. Tetraspanins are involved in sorting and targeting cargo to MVBs and, in cooperation with the ESCRT machinery, into EVs [92,93]. While it has been shown that HCMV-infected cells release EVs that contain viral surface proteins such as gB [94], the role of exosomal pathways in HCMV particle envelopment and release are broadly not defined. Although inhibitors of exosome biogenesis can slow HCMV spread, they do not significantly influence viral titers as shown here and by others [65,95], possibly arguing for an involvement of the MVB/exosome-pathways in cell-to-cell spread. Contradictory evidence exists for the role of CD63 in HCMV virus production. While one study did not find a significant effect of siRNA-mediated CD63 knock-down on HCMV titers [65], another recent study found a substantial reduction of HCMV titers upon CD63 siRNA knock-down [66]. The reason for this discrepancy is difficult to determine since the experimental settings in which each of the datasets was acquired varied drastically. This is especially true for the virus strains used in these studies. While Hashimoto *et al.*, as well as Turner et al., used the lab-adapted HCMV-AD169 strain, Streck *et al.* investigated the more clinical HCMV-TB40/E strain [46,65,66]. HCMV-AD169 is adapted to release large amounts of supernatant virus from *in vitro* cultured fibroblast, while clinical strains usually grow strongly cell associated [20,24,73,74]. The strain TB40/E, which resembles clinical isolates more closely than other lab-adapted strains, produces both cell-associated virus and cell-free virus [30,34,73]. It is, therefore, tempting to speculate that release pathways that rely on CD63 are used mainly by HCMV to produce cell-free virus, whereas other pathways responsible for cell-associated spread might not be impaired in the absence of CD63. However, the mechanisms and differences of cell-associated spread and spread through cell-free virus are still poorly understood and more data is needed.

In HFF cells infected with HCMV-TB40, we found colocalization between the tetraspanin CD63 and the viral envelope glycoproteins gB, gM, and the tegument protein pp150. However, CD81 and CD9, which are also associated with exosomes, did not colocalize with the viral markers as strongly. Since EVAs were negative for CD63, this marker might be excluded during the budding process at the MVB surface. However, this idea contradicts a previously published study showing that CD63 is incorporated in the virion envelope [88]. Importantly, we and others did not find significant enrichment of CD63 in proteomic analyses of purified TB40 virions [46]. CD63 possibly plays a role in the sorting of viral glycoproteins to sites of secondary envelopment, as tetraspanins are known to be involved in sorting plasma membrane-bound molecules into MVBs [92,93]. HCMV gB is known to localize to the plasma

membrane and be sorted through endocytic and recycling pathways by an acidic cluster in its cytoplasmic domain [96,97]. For HSV-1, it was reported that disrupting the endosome-to-MVB trafficking pathway leads to the mislocalization of HSV-1 gB [98]. More recently, it has also been shown that HSV-1 replication leads to an increase in the exocytosis of CD63-containing extracellular vesicles, leading the authors to hypothesize that HSV-1 modulates exosome biogenesis for its benefit [99]. Taken together, these reports indicate that endocytic pathways can be involved in the trafficking of viral factors to sites of herpesvirus secondary envelopment. Our observation that HCMV gB strongly localized with CD63 might support this hypothesis and fits a recent report that gB is enriched in exosomes [94]. Moreover, a recent proteomics study focusing on exosome release from HCMV infected cells aligns with this interpretation [46]. This study further identified several additional viral proteins that likewise appear in exosomes. The data provided by the authors strengthen the overall idea that HCMV exploits endocytic trafficking and exosome biogenesis pathways for the assembly and egress of virus particles. However, how much of the host factors involved with exosome generation in the absence of virus infection are involved in virus particle production remains unclear. In our hands, the MVB inhibitor U18666A did not influence virus production. In contrast, Tipifarnib, an inhibitor of exosome biogenesis, significantly reduced virus titers 4 dpi and EVA generation. Tipifarnib has been shown to reduce Rab27a, nSMase2, and Alix levels, which might result in an effect on trafficking of viral components to assembly sites or membrane remodeling during secondary envelopment [70]. Alternatively, inhibition of the cellular farnesyltransferase by Tipifarnib might also act on Ras signaling pathways, which have been reported to positively influence HCMV, HSV-1, and other herpesvirus infections [100–102]. Still, an effect of Tipifarnib on transcription would be the simplest explanation for the reduced pp150 levels at 3 and 4 dpi, which makes our data hard to interpret at this point. On the other hand, inhibition of downstream HCMV assembly processes might also result in the degradation of structural proteins such as pp150. Moreover, it is conceivable that the host farnesyltransferase is directly involved in the post-translational modification of virus proteins and that its inhibition by Tipifarnib has a negative effect on viral protein levels and replication.

Instead of being the result of an altered MVB pathway, MViBs might also originate from the fusion of individual virus-filled transport vesicles as described for the related betaherpesvirus HHV-6A [60]. This model fits reports that MViBs were mostly found in the AC periphery while most capsid budding into individual vesicles is observed in the center of the AC, where early endosomal markers and Golgi-markers merge [41,42]. However, in the work we present here, we regularly found potential budding events at MViBs but could not identify intracellular vesicle fusion events leading to MViB formation in entire 3D-EM volumes of infected cells. We, therefore, conclude that MViB-mediated HCMV egress is a novel spatio-temporally separated egress pathway.

The production of HCMV progeny and their virion properties is a complex problem, determined mostly by the genetic properties of the strain and the producer cell. As a result, depending on strain and infected cell type, HCMV can generate cell-associated and cell-free populations, which in turn can also have differences in infectivity and tropism [8,20,24,30,32,34,73,78,103]. Moreover, these different particle populations can vary significantly in their trimeric to pentameric glycoprotein complex composition, which seems to influence their cell tropism to a large degree [8–10,14,24,31,32,34,73,78,79,104]. It is plausible to hypothesize that these virus populations might also undergo different envelopment processes in the cell and are exocytosed with a different spatio-temporal profile as suggested before [29,34]. A recent study from the Wilson and Goodrum labs suggests that virus-containing MVBs in fibroblasts and endothelial cells are derived from different cellular membranes,

which would add another potential HCMV egress pathway that could result in different virus populations; however, it is unclear if these pathways are functional in egress [54].

Future work needs to focus on characterizing the particle populations exocytosed by these different pathways regarding their glycoprotein content and define their role in potentially divergent egress routes. For most of the experiments, we used the HCMV strain TB40, which has been suggested to produce two virus populations on HFF cells which are endothelial-cell and fibroblast-tropic [34]. The EVAs that we found were largely static during live-cell imaging and might represent a cell-associated viral population. We found EVAs not only trapped between the cell and the cell support but also on the upper side of infected cells, as well as between cells. This observation would support the idea of cell-to-cell spread. However, our proteomics data and a recent study [46] found that soluble, purified virions showed markers of the exosome pathway. If the virions released through EVAs are the only ones that carry exosome-markers, this would suggest that it is unlikely that they stay cell-associated and play a role in cell-to-cell spread.

In summary, our data, combined with published studies, suggest a model in which membranes originating from a fusion of both the endosomal and trans-Golgi network are used for either individual envelopment of capsids or to generate MViBs in two spatio-temporally separated processes. MViBs are then transported to the plasma membrane, where fusion results in bulk pulses of virus particle exocytosis and the formation of EVAs (Fig 7). Future work is

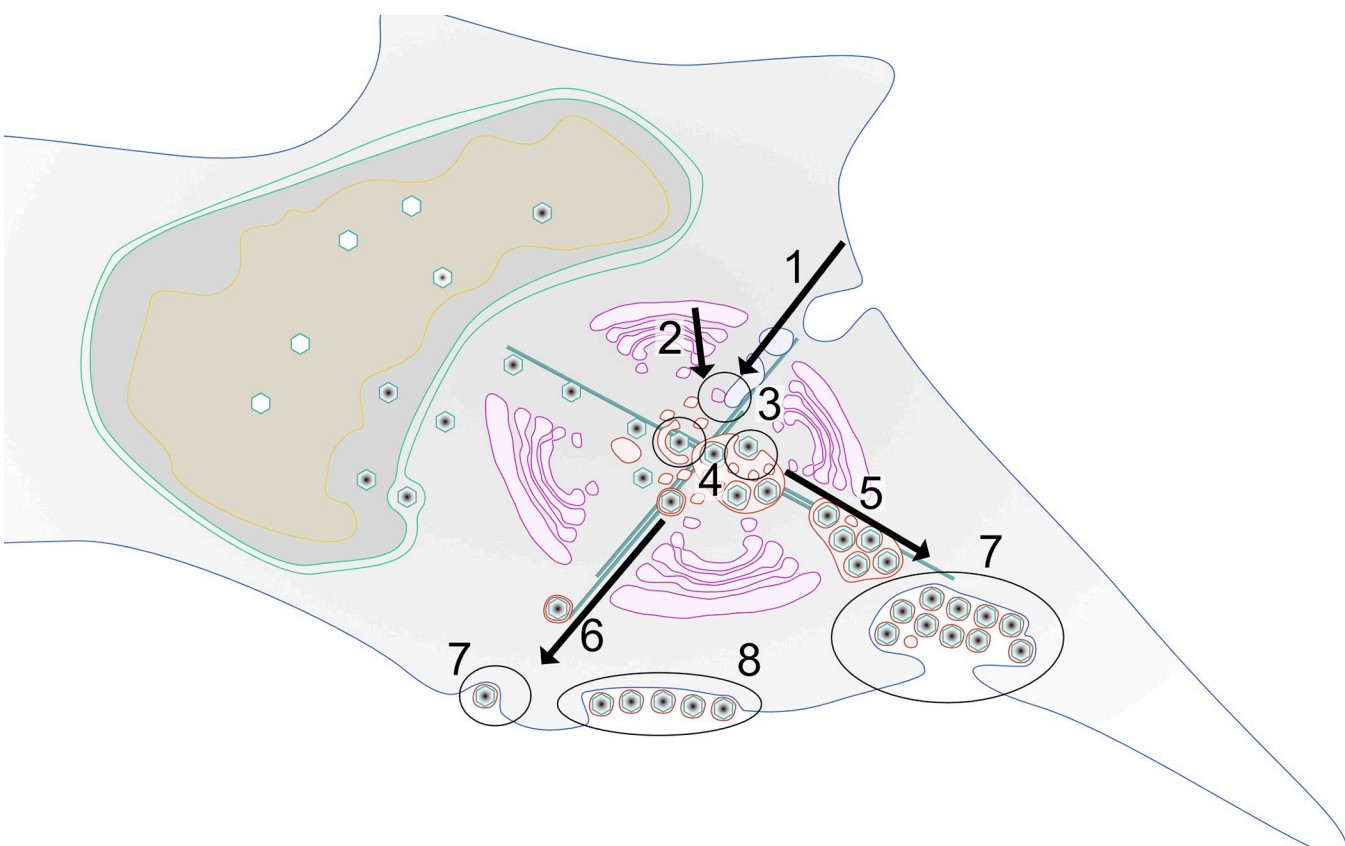

**Fig 7. Model of possible HCMV release pathways. 1–3** Membranes of late-endosomal and trans-Golgi origin are trafficked to the center of the assembly complex and subsequently utilized for secondary envelopment. **4** After egress from the nucleus, capsids are trafficked to the assembly complex where they either bud individually into single vesicles or into MViBs to acquire their membrane envelope. **5–6** Virus-containing vesicles and MViBs are transported towards the plasma membrane, and **7** fuse with it to release their content to the extracellular space. **8** MViB fusion leads to EVA formation.

needed to delineate the biogenesis of MViBs and, importantly, their potential role in producing specific virus populations.

# Materials and methods

## Cells and viruses

HFF-1 cells (ATCC-SCRC-1041, ATCC) were cultivated in Dulbecco's Modified Eagles Medium Glutamax (Thermo Fisher Scientific), supplemented with 5% FBS superior (Merck) and 5 μl of $10^6$ units/ml recombinant human FGF (PeproTech) / 500 ml Medium. HCMV-pp150-EGFP-gM-mCherry was a kind gift by Christian Sinzger [61]. The HCMV-TB40-BAC4 was a kind gift by Wolfram Brune [23]. HCMV-Merlin-pAL1502-WT and HCMV-Merlin-pAL1502-pp150-EGFP-gM-mCherry were a kind gift by Christian Sinzger [74]. For infection experiments, we used cell-free, infectious supernatant derived from HFFF-TetR cells that was a kind gift of Christian Sinzger. In these cells, RL13 and UL128 expression is suppressed as described by Murrell et al. 2017 [73] such that virions used for infection were lacking RL13 and UL128. Imaging was done on normal HFF-1 without the tet-repressor such that the analyzed cells expressed RL13 and UL128 [24,73].

Different multiplicities of infection (MOIs) were used for the infection experiments. In general, low MOI infections were used to avoid artifacts generated by high virus doses. Therefore, whenever possible, we used MOIs between 0.5 and 1. However, for particular experiments, such as bulk assays or electron microscopy, we used MOIs of up to 5. The used MOI is indicated for each experiment.

## Spinning-disk fluorescence microscopy

Spinning-disk microscopy was carried out on a Nikon TI2 (Nikon) based spinning-disk system equipped with a Yokogawa W2, a Nikon 1.49 NA Apo-TIRF objective, and an Andor iXON888 EMCCD (Andor Technology). The resulting pixel size was 130nm, and image acquisition was done with NIS-Elements. Further, the setup was equipped with 405, 488, 561, and 640 laser lines and corresponding filter sets. Live-cell experiments were carried out with a humidified incubation chamber heated to 37°C and 5% CO2 controlled by a gas mixer. For fluorescence microscopy, cells were grown in Ibidi 35mm glass-bottom dishes or Ibidi 8-well glass-bottom chamber slides (Ibidi GmbH), for CLEM in Ibidi 35mm grid polymer bottom dishes. SNAP labeling before live-cell imaging with SNAP-Cell 647-SIR (New England Biolabs GmbH) was done according to the manufacturer's instructions. Image processing and analysis were performed in ImageJ/FIJI.

## Serial Block Face Scanning Electron Microscopy (SBF-SEM)

For SBF-SEM, cells were fixed at the indicated time-points with 2% Paraformaldehyde (PFA/ Science Services) and 2.5% Glutaraldehyde (GA/ Science Services GmbH) in Dulbecco's phosphate-buffered saline (D-PBS, Sigma-Aldrich) for 5 minutes at room temperature (RT) and 55 minutes on ice. Subsequently, the sample was processed with the following procedure: Post-fixation with 2% Osmium Tetroxide (OsO4/ Science Services) and 2.5% GA in D-PBS on ice, staining with 2% $OsO_4$, 1.5% potassium ferrocyanide (Sigma-Aldrich), 2mM $CaCl_2$ (Sigma-Aldrich) in water, incubation in 0.5% thiocarbohydrazide (Sigma-Aldrich) in water, staining with 2% $OsO_4$ in water, incubation in 1% gallic acid (Sigma-Aldrich) in water, staining with 2% uranyl acetate (Merck KGaA) overnight in water. On the next day, the sample was stained with freshly prepared Waltons lead aspartate [105] ($Pb(NO_3)_2$ (Carl-Roth), L-Aspartate (Carl-Roth), KOH (Merck)), and subsequently subjected to a PLT dehydration series to Ethanol

Rotipuran (Carl-Roth). Finally, the samples were infiltrated with 70% Epon in Ethanol before two incubations with 100% Epon and the final polymerization was carried out in Epon supplemented with 3% silver flakes (Sigma-Aldrich) and 3% (w/w) Ketjen Black (TAAB). Sample blocks of 0.5x0.5 mm were cut, mounted, and inserted into a Gatan 3View stage (Gatan) built in a Jeol JSM-7100F scanning electron microscope (Jeol). For imaging, the sample stage was biased with a 500V positive charge to account for sample charging during the scanning process. For the acquisition, 3x3 nm pixel size images were scanned, followed by the repeated ablation of 50 nm sections. The acquisition was controlled by the Gatan Digital Micrograph software, which was also used for stack alignments. Further processing of the datasets was performed in FIJI, and the volumes were rendered in Imaris 8 (Bitplane). To quantify MViB and EVA compositions, subvolumes of those structures were randomly chosen and extracted. Subsequently, particles were manually identified and counted. The image handling tasks were performed in ImageJ/FIJI. Statistical analysis was performed in GraphPad Prism 8.

## Transmission Electron Microscopy (TEM)

For TEM, cells were fixed and processed as described for SBF-SEM up to the embedding step. The cells were embedded in Epon without fillers, sectioned to 50 nm on a Leica Ultracut Microtome (Leica), and transferred to copper mesh grids. Electron microscopy was performed on an FEI Tecnai G20 (FEI/ Thermo Fisher Scientific), and images were acquired on an Olympus Veleta side-mounted camera (Olympus). A simplified staining protocol was used for the TEM-CLEM experiments with the HCMV-Merlin-pAL1502 variants: Before spinning-disk microscopy, the infected cells were fixed with 4% PFA in PBS at 37˚C and 5% $CO_2$ for 10 min. Afterwards, cells were postfixed at 4˚C with 2.5% GA in PBS O/N and stained on ice using 1% $OsO_4$/PBS for 30 min and 2% $UA/H_2O$ for 30 min. Subsequently the sample was dehydrated through a series of increasing Ethanol concentrations in water (30%, 50%, 70% and 3x 100%; each 10 minutes on ice). Subsequently, the sample was infiltrated by increasing concentrations of Epon in Ethanol (50% for 30 min, 70% 1:30h, 100% O/N) before polymerization at 60˚C for 2 days. Sectioning was performed as described above. Before TEM, the sections were post-stained for 7 minutes with a saturated solution of UA in 70% Ethanol/$H_2O$.

## Lattice light sheet microscopy

Lattice light-sheet microscopy was performed on a Zeiss Lattice Light Sheet 7 (Carl Zeiss) as part of an early adoptor program, controlled with Zeiss Zen Blue software. The device is equipped with 488, 561, and 640 laser lines and multi-bandpass filters. Live-cell experiments were carried out on Ibidi 35mm glass-bottom dishes at 37˚C with 5% CO2 in a humidified atmosphere. Images were acquired on a pco.edge (PCO AG) sCMOS camera with a final pixel size of 145nm. Images were deconvolved after acquisition in Zen Blue using the built-in constrained-iterative algorithm. 2D image processing was done in Zen Blue, arrangements and montages were made in FIJI. 3D image processing was done in Arivis 4D (arivis AG); videos were cut and arranged in Adobe Premiere Pro (Adobe Inc).

## BAC mutagenesis

BAC mutagenesis was performed as described before by en-passant Red Recombination [106]. The creation of HCMV-TB40-BAC4-pp150-SNAP-gM-mScarlet-I was done in two steps. At first, UL32 (gene locus of pp150) was mutated by the C-terminal insertion (after K1045) of the SNAP-Tag-SCE-I-KanR shuttle sequence with a nine amino acid linker (HTEDPPVAT) and subsequent second recombination to clear the Kanamycin resistance and restore the SNAP-Tag sequence (NEB; for complete insertion sequence see Table 1). This was followed by the

**Table 1. Sequences for 2-Step BAC mutagenesis of HCMV-TB40-pp150-SNAP-gM-mScarlet-I.**

| TB40-pp150-SNAP Insert Sequence | |
|---|---|
| Insert Sequence | CACACGGAGGATCCACCGGTCGCCACC<br>atggacaaagactgcgaaatgaagcgcaccaccctg<br>gatagccctctgggcaagctggaactgtctgggtgcg<br>aacagggcctgcacgagatcaagctgctgggcaaag<br>gaacatctgccgccgacgccgtggaagtgcctgcccc<br>agccgccgtgctgggcggaccagagccactgatgca<br>ggccaccgcctggctcaacgcctactttcaccagcctg<br>aggccatcgaggagttccctgtgccagccctgcaccac<br>ccagtgttccagcaggagagctttacccgccaggtgct<br>gtggaaactgctgaaagtggtgaagttcggagaggtca<br>tcagctaccagcagctggccgccctggccggcaatcc<br>cgccgccaccgccgccgtgaaaaccgccctgagcgg<br>aaatcccgtgcccattctgatcccctgccaccgggtggt<br>gtctagctctggcgccgtggggggctacgagggcggg<br>ctcgccgtgaaagagtggctgctggcccacgagggcc<br>acagactgggcaagcctgggctgggt |
| **TB40-gM-mScarlet-I Primer (50bp overhangs)** | |
| Forward | ACT ATC ACG TCG TGG ACT TTG AAA GGC TCA ACA TGT CGG CCT ACA ACG TAG TGA GCA AGG GCG AGG C |
| Reverse | CAC ACC AGC TGC ACC GAG TCT AAG AAA AGC ATA GGC GTG TGC AGG TGC ATC TTG TAC AGC TCG TCC ATG CC |

insertion of the mScarlet-I [107] sequence in the UL100 gene between the codons for amino acids V62 and M63 of gM by amplifying the mScarlet-I-SCE-I-KanR shuttle construct with the primers shown in Table 1, with the second recombination as described for the first step. The virus was reconstituted by electroporation of the BAC DNA into HFF cells.

## Gateway cloning and lentivirus transduction

Plasmid pCMV-Sport6-CD63pHluorin was a gift from DM Pegtel through Addgene (Addgene plasmid # 130601; http://n2t.net/addgene:130901; RRID:Addgene_130901, Addgene). For Gateway (Thermo Fisher Scientific) cloning, the pCMV-Sport6-CD63pHluorin was recombined with pDONR-221 (Thermo Fisher Scientific) to produce the pENTR-CD63pHluorin vector that was further recombined with pLenti-CMV-Puro-DEST (w118-1), a gift from Eric Campeau & Paul Kaufman through Addgene (Addgene plasmid # 17452; http://n2t.net/addgene:17452; RRID: Addgene_17452).

The resulting pLenti-CMV-CD63pHluorin-Puro was then transfected with polyethyleneimine (Polysciences) together with 3rd generation Lentivirus vector helper plasmids, gifts by Didier Trono, RRID: Addgene_12253, Addgene_12251, Addgene_12259) into 293XT cells (Takara Holdings). Lentivirus containing supernatant was harvested at 48, 72, and 96 hours post-transfection, filtered through 0.2 µm syringe filters, and used to transduce HFF-1 cells. 72hpi, the HFF-cells were selected with Puromycin (Thermo Fisher Scientific) at 5 µg/ml. Furthermore, the cells were sorted by fluorescence-activation (FACS), using a FACS Aria Fusion (BD Biosciences), for the 10% strongest fluorescent cells, further cultivated and used for the experiments.

## Immunofluorescence

For immunofluorescence experiments, cells were grown in 35mm glass-bottom Ibidi dishes and fixed at the indicated time-points with 4%PFA in D-PBS. SNAP labeling with SNAP-Cell 647-SIR was done as described in the manual for SNAP-Cell 647-SIR (NEB). Afterward, the samples were permeabilized with TritonX-100 at 0.1% in D-PBS with subsequent blocking with 3% Bovine Serum Albumin (Sigma-Aldrich) in D-PBS. Primary antibodies used in this

study were Ultra-LEAF Purified anti-human CD63 H5C6 (Biolegend), Anti-Cytomegalovirus Glycoprotein B antibody [2F12] (ab6499) (Abcam), Purified anti-human CD9 HI9a (Biolegend), Purified anti-human CD81 (TAPA-1) 5A6 (Biolegend). Secondary antibodies used were Alexa 647 goat anti-mouse (Thermo Fisher Scientific) and Alexa 488 goat anti-mouse (Thermo Fisher Scientific).

## Quantification of the frequency of extracellular viral accumulations

HFF-WT cells were infected with HCMV-TB40-pp150-SNAP-gM-mScarlet-I or HCMV-TB40-WT at an MOI of 1 and fixed at 120hpi. HCMV-TB40-WT infected cells were stained for gB as described for the other immunofluorescence experiments. Late infected cells were identified in WT-infected cells by a well identifiable gB-positive assembly complex. Z-stack tilings were acquired by spinning-disk microscopy. In the HCMV-pp150-SNAP-gM-mScarlet-I infected cells, late infected cells were identified by three conditions: 1) Well identifiable gM-positive assembly complex. 2) Nuclear signal of pp150. 3) Significant pp150 signal in the assembly complex. Quantification of EVAs in HCMV-Merlin-WT and HCMV-Merlin-pp150-EGFP-gM-mCherry was carried out in the same way.

## Confocal scanning imaging

Confocal Laser Scanning Microscopy was carried out on a Nikon TI2 microscope equipped with an A1 confocal laser scanning unit, a 1.4 NA 60x Plan Apo objective, PMT, and GaAsP detectors, standard 404, 489, 561, and 637 laser lines, and corresponding filter sets (Nikon). Imaging conditions were optimized for each sample. Scan sizes were adapted to fulfill the criteria for Nyquist-sampling, resulting in a pixel size of 118 nm. The acquisition was run in NIS-Elements, post-processing and image analysis were performed in FIJI.

## Weighted spatial colocalization analysis

For the weighted colocalization heatmaps, pixel intensities were calculated, taking into account the absolute intensities in both channels and the ratio between the intensities. The calculation was performed by first normalizing each channel to relative intensity. In the following, the relative intensities of each pixel in both channels $a$ and $b$ were interpreted as a vector $\begin{pmatrix} a \\ b \end{pmatrix}$

Describing the vector to the position of that pixel in a classical scatter plot. The length of the vector was then multiplied by $1 - |\sin(\alpha) - \cos(\alpha)|$ while $\alpha$ is the angle between the vector and the x-axis. This multiplication emphasizes pixels where the two colors colocalize with similar relative intensities. The product then was plotted back to the original pixel position in the image resulting in the heatmap shown in the figures. With this strategy, we could put the information of a 2-channel scatter plot back into the image's spatial context.

The Jupyter notebook for this analysis is available on GitHub:
(https://github.com/QuantitativeVirology/2D-Colocalization)

## Gradient purification of HCMV

A 15 cm dish of HFF cells was infected with HCMV-TB40-WT at MOI 0.05. Seven dpi, the infected cells were trypsinized and split onto 16x 15 cm dishes of HFF cells. 7 days after subculturing, the supernatant was harvested and clarified by centrifugation at 1200 xg for 5 min. The virus was pelleted by centrifugation at 14000xg for 1.5 h at 4˚C and then resuspended in 1% FBS/PBS overnight on ice. The resuspended virus was centrifuged at 18000xg for 1 min at 4˚C to remove large aggregates and then loaded over a continuous gradient made from 15%

sodium tartrate with 30% glycerol (w/w) and 35% sodium tartrate (w/w) in 40 mM sodium phosphate pH 7.4 [108]. The gradient was made with a Gradient Master (BioComp Instruments) for an SW41 rotor. After centrifugation at 65000xg for 1.5 h at 4˚C, the bands were isolated, diluted 10-fold in PBS, and pelleted at 14000xg for 1.5 h at 4˚C. The purified virus pellet was resuspended overnight in PBS and stored at -80˚C.

## Mass spectrometry

The purified virus was mixed with 3 volumes of lysis buffer (100 mM Tris, 50 mM DTT, 8 M Urea pH 8.5) and incubated at room temperature for 15 min. Samples were digested using the FASP protocol, employing 30 kDa molecular weight cut-off filter centrifugal units (Amicon, Merck, [109]). Briefly, the lysed virus was added to the centrifugal unit and washed with TU buffer (100 mM Tris, 8 M Urea pH 8.5). Next, 8 mM DTT in TU buffer was added and incubated at 56˚C for 15 min. After two further washes, 50 mM iodoacetamide (IAA) in TU buffer was added and incubated for 10 minutes at room temperature. The centrifugal units were washed twice, treated again with DTT, washed once further with TU buffer, and twice with 50 mM ammonium bicarbonate solution. MS grade trypsin (Promega) was added in a 1:100 enzyme:protein ratio, and the sample was incubated overnight at 37˚C. The flow-through containing trypsinized peptides was collected and pooled, and the sample was lyophilized with a SpeedVac (Thermo Fisher Scientific). The resulting peptides were enriched with C18 stage tips prepared in-house and eluted with 80% acetonitrile containing 0.5% acetic acid. The samples were dried down by SpeedVac (Thermo Fisher Scientific) and resuspended in 97% water, 3% acetonitrile with 0.1% formic acid, and 10 fmol/μL E. coli digest (Waters Corporation) for analysis by LC-MS/MS.

Peptides resulting from trypsinization were analyzed on a Synapt G2-Si QToF mass spectrometer connected to a NanoAcquity Ultra Performance UPLC system (both Waters Corporation). The data acquisition mode used was mobility enhanced MSE over m/z range 50–2000 with the high energy collisional voltage in the transfer region ramped from 25 to 55 V. Mobile phases used for chromatographic separation were water with 0.1% formic acid (A) and acetonitrile with 0.1% formic acid (B). Samples were desalted using a reverse-phase SYMMETRY C18 trap column (100 Å, 5 μm, 180 μm x 20 mm, Waters Corporation) at a flow rate of 8 μl/min for 2 minutes. Peptides were separated by a linear gradient (0.3 μl/min, 35˚C; 97–60% mobile phase A over 90 minutes) using an Acquity UPLC M-Class Reversed-Phase (1.7 μm Spherical Hybrid, 76 μm x 250 mm, Waters Corporation).

LC-MS data were peak detected and aligned by Progenesis QI for proteomics (Waters Corporation). Proteins were identified by searching against the Human and HCMV proteomes in Uniprot. The database search was performed with the following parameters: mass tolerance was set to software automatic values; enzyme specified as trypsin; up to two missed cleavages; cysteine carbamidomethylation as a fixed modification, with the oxidation of methionine, S/T phosphorylation, and N-terminal acetylation set as variable modifications. Abundances were estimated by Hi3-based quantitation [110].

For comparison with the Turner et al. (2020; [46]), dataset, protein accession was converted to UniParc codes.

## Live TIRF microscopy

For live-cell TIRF imaging, infection experiments were carried out in 35 mm glass-bottom Ibidi dishes. SNAP labeling with SNAP-Cell 647-SIR was done as described in the manual for SNAP-Cell 647-SIR (NEB) before imaging. Microscopy was performed on a Nikon TI equipped for TIRF microscopy and equipped with standard 488, 561, and 640 laser lines,

corresponding filter sets, and an incubation chamber with a heating system. The illumination angle was determined experimentally by manually adjusting for TIRF illumination, and image acquisition was performed with NIS-Elements using an ANDOR iXon Ultra 897 EMCCD camera. Live-cell experiments were carried out at 37˚C. Intensity measurements in the time courses were done with FIJI by manually placing ROIs. The data analysis and visualization in the graphs were performed in GraphPad Prism 8.

## Immunogold labeling

For immunogold labeling of HCMV infected cells, 1x 10 cm cell culture dish of HFF-WT cells were infected with HCMV-TB40-WT at an MOI of 0.5. At 4 dpi, the cells were fixed with a mixture of 2% PFA and 0.5% GA (both Science Services) in PBS (Sigma-Aldrich) for 10 minutes at 37˚C and 5% $CO_2$. The cells were washed once in PBS and subsequently scraped in 1% gelatin (food grade brand). The cells were pelleted, resuspended in 10% gelatin, and pelleted again while letting the gelatin cool to solidify. The gelatin with the embedded cell pellet was cut into small (1–3 mm) chunks, immersed in 2.3M Sucrose solution, and stored overnight at 4˚C. The next day, the pieces were mounted on a sample holder and flash-frozen by immersion in liquid nitrogen. Afterward, the pieces were trimmed and sliced into 70 nm thin sections on a Leica EM FC7 cryo-microtome (Leica) using a diamond knife (Diatome). The sections were recovered by picking them up with a drop of 2.3M Sucrose and transferring them to Formvar and carbon-coated nickel grids, letting the sections thaw in the process. In the following, the sections were immunogold labeled by the following protocol: removal of residual gelatin by incubation in PBS for 20 minutes at 40˚C. 3x 2 minutes incubation with PBS, 3x 2 minutes incubation in 0.1% Glycine (Sigma-Aldrich) in PBS, and blocking for 3 minutes in Aurion donkey serum (Aurion/ Science Services). After blocking, the samples were incubated with Ultra-LEAF Purified anti-human CD63 H5C6 antibody (Biolegend), diluted 1:5 in Aurion donkey serum, for 30 minutes. Afterward, the samples were washed 5x 2 minutes in PBS and incubated with 10 nm gold coupled donkey-anti-mouse IgG (Aurion/ Science Services), diluted 1:20 in Aurion donkey serum, for 1 hour. Subsequently, the samples were washed 5x 2 minutes in PBS, followed by fixation with 1% GA in PBS for 5 minutes. Afterward, the samples were incubated 10x 1 minute in distilled water and stained, first with uranyl acetate (Merck) in water and secondly with uranyl acetate in 1% methylcellulose. Finally, the grids were air-dried after blotting the uranyl acetate-methylcellulose solution and observed by transmission electron microscopy.

## Inhibitor treatments

U18666A was acquired from Merck, Ketotifen-fumarate, and Tipifarnib were bought from Sigma-Aldrich. The substances were dissolved in DMSO to produce stock solutions (U18666A at 4 mg/ml, Ketotifen at 20 mM, and Tipifarnib at 5 mM), which were subsequently aliquoted and frozen at -80˚C. The drugs were added to the complete growth medium at the indicated time points and at the concentration indicated for each experiment. The medium containing the inhibitor was renewed every 24 hours.

## Titrations

To assess the virus titers in supernatant from HCMV infected cells. HFF-WT cells were seeded in 24-well dishes to reach 90–100% on the next day. Tenfold dilutions of the harvested infectious supernatants were made in complete growth medium from $10^{-1}$–$10^{-4}$. The medium from the HFF cells was removed and replaced by 100 µl of one the dilutions per well. The plates were rocked gently every 15 minutes for 1 hour to ensure even distribution. After one

hour of incubation, an overlay of DMEM containing 2% FCS and 0.6% Methylcellulose (Sigma-Aldrich) was added to the cells, and the plates were incubated at 37˚C and 5% $CO_2$ for 14 days. Afterward, the cells were fixed, and fluorescent virus plaques were counted. For non-fluorescent virus variants, the samples were stained against HCMV-IE1/2 with anti-IE1/2 hybridoma supernatant ([111], kind gift by Wolfram Brune) analogous to the procedure described in the immunofluorescence section.

## Cytotoxicity assays

HFF-WT cells were seeded on a black 96-well plate to reach confluency on the next day. Then the cells were treated with the indicated substance and concentration for 24 hours. Afterward, the cell viability was measured using the CellTiter-Glo luminescent cell viability assay (Promega) and a FLUOStar Omega plate reader (BGM Labtech), both according to the instructions from the manufacturer.

## Western blotting

HFF-WT cells were infected with HCMV-TB40-WT (MOI = 3), cells were harvested and lysed at 1 hpi (input) and every 24 hours until 96 hpi. SDS-PAGE was performed on Bio-Rad Mini-PROTEAN TGX 4–15% gels (Bio-Rad). Separated protein was blotted on Amersham Protran 0.45 µm nitrocellulose membranes (Cytiva). The membrane was cut, and the sections were subsequently stained with one of the primary antibodies against HCMV pp150 (kind gift by Eva-Maria Borst and Stipan Jonjic), anti-CMV ICP36 monoclonal antibody 10D8 (Virusys), and anti-IE1/2 (hybridoma supernatant [111], kind gift by Wolfram Brune) followed by a secondary antibody Goat Anti-Mouse IgG StarBright Blue 700 (Bio-Rad). An Rhodamine coupled Anti-GAPDH hFAB antibody (Bio-Rad) was used as loading control. The stained blots were imaged using a ChemiDoc MP imager (Bio-Rad).

## Data deposition

The experimental data was uploaded to Dryad and PRIDE [112] online repositories. Mass spectrometry data was uploaded to PRIDE [113]. The other data was split due to size constraints. SBFSEM data was deposited separately in [114] and the remaining data is in a combined dataset [115].

## Dryad DOI

https://doi.org/10.5061/dryad.gtht76hpt [115]

## Supporting information

**S1 Fig. SBF-SEM can visualize all steps of HCMV virus particle morphogenesis.** HFF cells were infected with HCMV-pp150-EGFP-gM-mCherry (MOI 3) and processed for EM 4 dpi. Image signals were inverted to facilitate comparison with TEM images. N marks nucleoplasm, C indicates cytoplasm, and Ex the extracellular space. Highlighted in the panels are examples of B-capsids in the nucleus (unfilled black triangles), DNA- filled nuclear C-capsids (white triangles with black contour), cytoplasmic non-enveloped C-capsids (black triangles), intracellular, enveloped virus particles (black filled arrowheads) as well as enveloped, released particles (empty arrowhead with black contour). Scale bar lengths are specified in each image. (TIF)

**S2 Fig. Potential capsid budding-event at MViB in relation to the infected cell architecture. S2A** Single SBF-SEM section of an infected HFF cell. HFF cell infected with HCMV-

pp150-EGFP-gM-mCherry (MOI 3) at 4dpi. Image signals were inverted to facilitate comparison with TEM images. The white frame indicates the area cropped and enlarged in B, showing the surface of an MViB in the periphery of the assembly complex. Scale bar indicates 10 μm. **S2B** A detail showing a single particle potentially budding into an MViB (white arrowhead). Scale bar indicates 200nm.
(TIF)

**S3 Fig. Overview of an HCMV assembly complex in an infected HFF cell.** HFF cell infected with HCMV-pp150-EGFP-gM-mCherry (MOI 3) at 4dpi. Shown is the assembly complex in a resliced section through an SBF-SEM stack. Scale bar indicates 1.5 μm. Image signals were inverted to facilitate comparison with TEM images.
(TIF)

**S4 Fig. Virus particle budding into an individual small vesicle. S4A** Single SBF-SEM section of an infected HFF cell. HFF cell infected with HCMV-pp150-EGFP-gM-mCherry (MOI 3) at 4dpi. The white frame indicates the area cropped and enlarged in B. Scale bar indicates 10 μm. **S4B** A detail showing a single capsid budding into a single vesicle (white arrowhead). Scale bar indicates 200 nm. Image signals were inverted to facilitate comparison with TEM images.
(TIF)

**S5 Fig. Tetraspanin CD63 colocalizes with gM and gB. S5A** HFF-CD63-pHluorin cells were infected at an MOI of 1 with HCMV-pp150-SNAP-gM-mScarlet-I. Cells were fixed at 4 dpi and stained for gB. The images show a representative cell and the localization pattern of the cellular MVB marker CD63 in relation to the viral glycoproteins gB and gM. CD63 localizes to large vesicles positive for gB and gM. Scale bars indicate 10 μm. The green line indicates the section quantified in S5B. **S5B** Line plot for the indicated areas in S5A. CD63 signal correlates with gM and gB signals in two MViBs. **S5C** Spatial weighted colocalization analysis highlights the specific areas for CD63 and gB colocalization. Scale bar indicates 10 μm.
(TIF)

**S6 Fig. Pathway analysis of virion mass spectrometry.** Pathway analysis of the mass spectrometry data from purified virions, done with string-db.org. The color of the dots indicates factors from either GO-term associated pathways or publications related to their functionality. The colors of the connections indicate the type of evidence for the interactions and are filtered for the highest interaction confidence (0.900) as provided by the database.
(TIF)

**S7 Fig. Tetraspanins CD9 and CD81 localize to the assembly compartment but not MViBs. S7A-B** HFF cells were infected at an MOI of 1 with HCMV-pp150-SNAP-gM-mScarlet-I. Cells were fixed at 4 dpi and stained with specific antibodies for CD9 (αCD9) and CD81 (αCD81). The images show representative cells and the localization pattern of the CD molecules relative to gM (gM-mScarlet-I) and pp150 (pp150-SNAP). Scale bars indicate 10 μm.
(TIF)

**S8 Fig. Virus products localize to large vesicles positive for CD63.** HFF cells were infected with HCMV-TB40-WT at an MOI of 0.5. After 4 dpi, the cells were fixed and processed for immunogold labeling against CD63. Note that membranes appear white in this preparation method, and low GA concentrations used to preserve epitopes might lead to less preservation of MViB contents. **S8A-G** Shown is large bodies containing virus particles (black triangles), dense bodies (white-filled triangles), and 10 nm gold particles (arrowheads). **S8H** A body with the classical phenotype of an MVB decorated with 10 nm gold particles (arrowheads). All scale

bars indicate 0.2 μm.
(TIF)

**S9 Fig. S9A and S9B HFF cells were infected with HCMV-TB40-pp150-SNAP-gM-mScar-let-I at an MOI of 2 and treated with the indicated substance at the indicated concentrations until 5 dpi.** The medium containing the inhibitors was refreshed every 24 hours. 5 dpi cells were fixed, labeled with SNAP-Cell-SiR, and imaged by spinning-disk microscopy. White triangles indicate EVAs. Scale bars indicate 20 μm. **S9C** HFF cells were infected with HCMV-TB40-pp150-SNAP-gM-mScarlet-I at an MOI of 2 and treated with the indicated inhibitors at the indicated concentrations until 4 dpi. The medium containing the inhibitors was refreshed every 24 hours. At 4 dpi, the supernatant was collected and titrated on HFF cells. Bars show mean, and error bars indicate standard deviation. Statistical significance was probed using one-way ANOVA (p-values: Tipifarnib: <0.0001, U18666A: 0.4154, Ketotifen: 0.8364) and Dunnet's multiple comparisons tests (shown in the figure). **S9D** HFF cells were treated as described for S9 A-B. Large overviews were created by spinning-disk microscopy, and EVAs were quantified. Bars show mean, and error bars indicate standard deviation. Statistical significance was calculated using a one-way ANOVA (p = 0.0179; in total, 687 late infected cells from triplicates were counted) and Dunnet's multiple comparisons test (shown in the figure). **S9E** HFF were treated with the indicated substance at the indicated concentration for 24 hours. Cell viability was measured with an ATP assay after 24 h. The apoptosis inducer Staurosporine was used as a positive control. Bars show mean, and error bars indicate standard deviation. Statistical analysis by a 2-way ANOVA confirmed statistically significant differences in the viabilities of the three groups (p-value < 0.0001). The cytotoxicity of Tipifarnib was not significantly different from the vehicle control, as determined by Tukey's multiple comparisons test. In contrast, the change in cell viability of Staurosporin was significant in the same analysis. **S9F** Western blot of HFF cells infected with HCMV-TB40-WT (MOI = 3) and treated with 1 μM Tipifarnib or DMSO (0.01%; vehicle control). At each indicated time point (input = 1 hpi), the cells were harvested and lysed, and the blot was probed for pp150 as a late protein, IE1/2 as immediate-early gene products, or UL44 as an early gene. GAPDH served as the loading control.
(TIF)

**S10 Fig. S10A and S10B HCMV-Merlin-pAL1502 produces large cytoplasmic bodies.** HFF-cells were infected as described in Fig 6B and 6C. Large cytoplasmic vesicles positive for viral proteins (white arrowheads) could be found in infection with both Merlin-WT **(S10A)** as well as Merlin-pp150-EGFP-gM-mCherry **(S10B)**.
(TIF)

**S11 Fig. S11A and S11B HCMV-Merlin-pAL1502 produces MViBs.** HFF-cells were infected as described in Fig 6D and 6E. CLEM reveals large cytoplasmic vesicles positive for viral proteins to be MViBs.
(TIF)

**S1 Table. Mass Spectrometry Results.**
(XLSX)

**S1 Text. Legends for supplementary videos, which are available for download through Dryad [115].**
(DOCX)

## Acknowledgments

We thank Wolfram Brune, Christian Sinzger, and Kerstin Sampaio for their generous gift of reagents and the viruses HCMV-TB40-BAC4, HCMV-TB40-pp150-EGFP-gM-mCherry, HCMV-Merlin-pAL1502-WT and HCMV-Merlin-pAL1502-pp150-EGFP-gM-mCherry as well as their continuous support. We thank Zeiss for including us in their lattice light-sheet early adoptor program.

## Author Contributions

**Conceptualization:** Felix J. Flomm, Timothy K. Soh, Hannah M. Britt, Rudolph Reimer, Kay Grünewald, Jens B. Bosse.

**Data curation:** Felix J. Flomm, Timothy K. Soh, Linda Wedemann, Hannah M. Britt.

**Formal analysis:** Felix J. Flomm, Timothy K. Soh, Hannah M. Britt.

**Funding acquisition:** Felix J. Flomm, Konstantinos Thalassinos, Kay Grünewald, Jens B. Bosse.

**Investigation:** Felix J. Flomm, Timothy K. Soh, Linda Wedemann, Hannah M. Britt.

**Methodology:** Felix J. Flomm, Timothy K. Soh, Carola Schneider, Hannah M. Britt, Konstantinos Thalassinos, Søren Pfitzner, Rudolph Reimer, Jens B. Bosse.

**Project administration:** Konstantinos Thalassinos, Kay Grünewald, Jens B. Bosse.

**Resources:** Konstantinos Thalassinos, Rudolph Reimer, Kay Grünewald, Jens B. Bosse.

**Software:** Felix J. Flomm.

**Supervision:** Timothy K. Soh, Carola Schneider, Konstantinos Thalassinos, Rudolph Reimer, Kay Grünewald, Jens B. Bosse.

**Visualization:** Felix J. Flomm, Timothy K. Soh.

**Writing – original draft:** Felix J. Flomm, Jens B. Bosse.

**Writing – review & editing:** Felix J. Flomm, Timothy K. Soh, Linda Wedemann, Rudolph Reimer, Kay Grünewald, Jens B. Bosse.

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
