## [Decision Letter · Decision Letter 0]

28 Feb 2022

Dear Dr. Bosse,

Thank you very much for submitting your manuscript "Intermittent Bulk Release of Human Cytomegalovirus" for consideration at PLOS Pathogens. As with all papers reviewed by the journal, your manuscript was reviewed by members of the editorial board and by several independent reviewers. In light of the reviews (below this email), we would like to invite the resubmission of a significantly-revised version that takes into account the reviewers' comments.

As you may note, I am not the original handling editor that was assigned to this manuscript upon initial submission and I was not privy to the initial reviews. That being said, I secured reviews from three experts in the Herpesvirology field with specific focus on viral assembly and egress. All reviewers were in agreement that this is a cutting edge approach to evaluating how HCMV assembles and exits infected cells. All reviewers were highly supportive of the work and appreciated the modifications that were made to the manuscript. However, one common concern of the reviewers were the over dependence of results with a single strain of HCMV that may have evolved to produce high levels of cell free virus in tissue culture. It is their opinion (and my own) that the inclusion of a different strain of HCMV as a validation would greatly strengthen the work and would result in a more favorable view of its acceptance in PloS Pathogens. I know that this is not the response that you were hoping for but I am confident that the inclusion of this data is not to prohibitive and can be done in a timely fashion.

We cannot make any decision about publication until we have seen the revised manuscript and your response to the reviewers' comments. Your revised manuscript is also likely to be sent to reviewers for further evaluation.

Sincerely,

Eain A Murphy, Ph.D.

Associate Editor

PLOS Pathogens

Shou-Jiang Gao

Section Editor

PLOS Pathogens

Kasturi Haldar

Editor-in-Chief

PLOS Pathogens

orcid.org/0000-0001-5065-158X

Michael Malim

Editor-in-Chief

PLOS Pathogens

orcid.org/0000-0002-7699-2064

Reviewer's Responses to Questions

**Part I - Summary**

Reviewer #1: The manuscript by Flomm et al describes studies using sophisticated fluorescence and electron microscopy imaging techniques to characterize structures within HCMV-infected cells associated with virion morphogenesis and egress. The authors describe large collections of virus particles at the exterior periphery of late-state infected cells, which they term extracellular viral accumulations (EVAs). They also describe vesicles within the cells containing wildly varying numbers of enveloped viral particles (virions and dense bodies), which they term multiviral bodies (MViB). Imaging data appear to show that MViB are a site of secondary envelopment and that fusion between the limiting membrane of the MViB and the cellular plasma membrane releases enveloped viral particles in bunches (i.e. EVAs). The identification of host cell secretory proteins associated with MViBs is consistent with the general herpesvirus model of secondary envelopment/morphogenesis at compartments derived from the cell secretory pathway.

Overall, the data are of high quality and the interpretations seem reasonable. However, I come away with the sense of not having learned much about HCMV egress beyond; 1) seeing a more detailed picture of the morphologic range of secondary envelopment compartments, and 2) seeing that these compartments can be larger and contain more virions than is commonly depicted in schematic diagrams of the “assembly compartment” (AC). The authors seem to draw a distinction between the MViB and the traditional description of the AC. But, given the drastic remodeling of the secretory/endosomal compartments in HCMV-infected cells, and the apparently wide variation in the size and viral contents of the so-called MViBs, it could be that the previously named ACs might be a type of MViB. Are the endosomal/exosomal pathway markers associated with MViBs specifically not associated with the traditional “AC”? That said, these studies lay an excellent technical framework for the study the specific mechanism of viral and host cell factors in virion morphogenesis and egress. – Brent Ryckman, University of Montana.

Reviewer #2: Manuscript: Intermittent Bulk Release of Human Cytomegalovirus (PPathogens –D-22-00174)

Authors: Flomm, F., Soh, TK, Schneider, C., et.al.

This revised manuscript provides evidence from several sophisticated imaging technologies that human cytomegalovirus (HCMV) assembles in large cytoplasmic structures (MViBs) and infectious virus in these structures is release following an exocytosis and fusion with the PM. These authors argue this route of virus release opens an area of research to further define mechanisms of virus diversity. The version I received had been previously reviewed and had extensive changes in the text in response to the previous review.

The data provided in this paper clearly supports the hypothesis on line 180 but does not provide definitive data for this mechanism of egress. The lack of either loss or gain of function studies to test this hypothesis are a major limitation of this study because it leaves the reader with a novel mechanism for HCMV egress but no definitive conclusion. This is probably best summarized by the honest assessment by the authors of their data (lines 288-292). Finally, this study has no data that in any way would connect the interpretations of the imaging data with the release of virions with specific cell tropism. This may be a part of their hypothesis but it is not clear how this can be integrated into the data or discussion of the data that was presented in this manuscript.

Specific queries:

1) pp150 is associated with the capsid but it is not a capsid protein and can be found in infected cells without any spatial associations with capsids or capsid proteins (line 149).

2) Line 204. It is unclear what “relaxed into patches” means to this reviewer. I can understand a spatial accumulation but “relaxing” is not a very clear descriptor.

3) The resolution of some of the images that rely on the fluorescent protein tagged virus was such that the reader is really dependent on where the authors put their arrow in the figures. This could be improved so that the reader can evaluate the data without total reliance on the placement of an arrow.

4) The term of MViB is an obvious compromise. However, what endomembranes contribute to these structures? Can they be isolated and analyzed? This question is central to their hypothesis and as a result, this manuscript could be viewed as a preliminary report.

Reviewer #3: This study uses advanced microscopy and biophysics based approaches, e.g., live-cell lattice light-sheet microscopy (LLSM) and three-dimensional correlative light and electron microscopy (3D-CLEM), to study how HCMV virions bud from infected cells during late morphogenesis and exit. In response to previous reviewers, the authors stress that they interpret the formation of what they call multi-viral bodies (MViBs), avoiding any direct claims that these are "multivesicular bodies" (MVBs). Overall, their data support a "bulk release" model wherein large vesiclular structures containing many virions fuse with the plasma membrane to release many virions at once into the extracellular space.

The imaging methods are cutting edge and provide a new level of visual detail by which herpesvirus researchers can appreciate the dynamic process of HCMV egress from fibroblasts. That said, a weakness of the paper is that not much is really done to perturb any of the key cell biological processes in order to test a specific hypothesis for how so called MViBs are formed and how egress works. On a related note, the work observes the process in a single cell type-- fibroblasts, with a single HCMV strain TB40/E. Although it is remarkable that bulk release may well account for the predominate mode of HCMV virion egress from the infected cell, this observation appears to be the core central finding of the paper and no efforts are made here to determine in a comparative way how representative the cell biology of egress studied here might or might not apply more generally to naturally circulating HCMV strains, nor even to address other laboratory adapted isolates. Also, when the authors write they do not know if the MViBs "might be a degradation pathway," seems like a bit of a forced "straw man" argument used to frame their work as hypothesis based, when in fact some of the outcomes were likely obvious to begin with.

* Please note that I was not a reviewer of the original submission of this manuscript. I am providing my initial views on the current submission.

**Part II – Major Issues: Key Experiments Required for Acceptance**

Reviewer #1: I confess that I am not familiar enough with advanced imaging techniques to make technical comments. I am convinced that the structures described actually exist. Beyond my sense that these MViBs might represent a more complete characterization of the traditionally conceived “AC”, I am concerned that these studies make use of a single strain of HCMV. In several places the authors note the dramatic differences between HCMV strains in terms of envelope composition and modes of spread, and seem to offer these as important to the significance of their studies. I agree that differences in the assembly/egress compartment might go a long way to explaining strain variation. Therefore, it is disappointing that the studies involve exclusively TB40-BAC4. I understand that these are technically difficult experiments, in part because of the need for a dual fluorescence markers. I am reluctant to suggest that they must include other strains prior to publication. But a more comprehensive and accurate discussion of the literature as pertains to inter-strain variation would seem like a reasonable request. Addressing some of my “minor points” would be a good place to start.

Reviewer #2: See above

Reviewer #3: Fig 4 and Lines 213-242: the CD63-pHluorin system is well described in the text. However the data presented do not match the narration--which leads a reader to believe that a "flash" will be observed from the CD63-pHluorin channel when a MViB fuses with the plasma membrane to release viral cargo to the extracellular space. Neither SI Figure 8 or the Figure 4 panels themselves reveal any "flash" .. and no live cell imaging is shown that supports this claim, either. To be clear, this reviewer is not skeptical of the claims or model. But I do not find that the data presented support the claim.

It would be important to show this "bulk release" phenomena occurs in more than one cell type and for different strains. Right now all the data are from TB40/E in fibroblasts. How does this look for strains like BAC cloned repaired to fully clinical status Merlin or TR3 strains? TB40/E is likely to have adapted to cell culture in a manner that promotes high level release of cell free virions, so it is uncertain whether the data shown here are representative of "fully clinical" naturally circulating HCMV strains.

The authors also raise points about genes like UL135 which they argue might be important for the genesis of virus filled MVB like structures. Why not test some of these ideas? It seems unlikely for instance that AD169 would fail to generate them, even though it lacks UL135 and many other ULb' genes.. Overall the study would be more valuable if the authors showed or at least tested if some viral genes might be important for driving bulk release versus other egress modalities.

**Part III – Minor Issues: Editorial and Data Presentation Modifications**

Reviewer #1: • Lines 62-67. I am concerned that these statements concerning the trimeric and pentameric complex are simplified to the point of conveying erroneous information.

o Yes, it seems that cell-to-cell spread can be mediated by gH/gL/UL128-131 in the absence of gH/gL/gO; this applies to fibroblasts as well as epithelial/endothelial cells (see Laib Sampiao 2016) But, note that efficient infection of epithelial and endothelial cells by extracellular virus requires both gH/gL/UL128-131 and gH/gL/gO. (there are numerous examples of papers showing this, including Wille 2010, Zhou 2013, Kabanova 2016 and Stegmann 2017 JVI).

o Yes, it is clear that strains can vary drastically in the amounts of the gH/gL complexes, but the genetic correlates are less well understood than the statement here seems to suggest. Yes, UL148 and US16 seem to play roles in assembly and trafficking of gH/gL complexes, but this does not explain in the least the dramatic strain variance that we and others have noted. Murrell 2013 describes a G-T substitution in the UL128 of TB40/e BAC4 that affects mRNA splicing. They offer this as an explanation for the low gH/gL/UL128-131 in TB40 virions compared to Merlin BAC, but as we have noted, TR BAC is also low in gH/gL/UL128-131 but is like Merlin at this nt residue. So, this is not an adequate explanation either.

o The notion that a culture of cells infected with genetically pure isolate of HCMV releases particles that vary in the abundance of gH/gL complexes is interesting and perhaps even likely, but I’m not sure there is any clear data published that demonstrates this. Yes, the Scriviano 20212 paper concluded that there was heterogeneity in the amounts of gH/gL/Ul128-131 in the virion envelope. But there were no direct data showing this, and there are other interpretations for their observations. Moreover, the flow cytometry analyses of Vlasak 2016 seem to show a good deal of regularity in the amounts of gH/gL/UL128-131 among virions within a stock. That said, Li JVI 1995 suggested virion heterogeneity as an explanation for their observations of a reversible antibody-resistance phenotype that did not involve genetic changes to the antigen.

o Overall, I caution the authors with this kind of oversimplification as it may well detract from the significance of their work.

• Line 108; What makes this view “unbiased?” I suspect that some readers (including me) might object to the notion that any “view” can be “unbiased.” One of the deepest insights of the quantum revolution is that it is not possible to observe anything without imputing something during the act of observation.

• Fig 2G. How do you know this particle is in the act of budding/envelopment, and not “back fusing” or simply pushed into the vesicle membrane? Yes, I see the membrane curving around this particle, and that would be consistent with the model of budding, but there seems to be no way to know the fate of the structure in a static image as this.

• Fig 2E/Line 179-181; To suggest that the MViBs are the predecessor/source of EVAs based on “no significant difference” in content, seems weak given the next statement, “very heterogeneous in size and content.” If the later is true it would seem that that MViBs and EVAs would be similar in content regardless of their relationship; i.e., both are heterogeneous in size and contents.

• Line 181. Related to the above point, if some MViBs contain only a few particles and others several hundred, what is the authors’ definition of MViB? And why is this necessarily different than the “AC”?

• Fig 3/Video 5. Indeed these images are neat, but I have a hard time understanding how they are interpreted as the authors’ suggest; i.e. “relaxes at it”. I admit that this might be an instance that I do not have a trained eye for such imaging techniques.

• Fig 3C/Vid6-7; I accept that the authors’ can distinguish various structures in the cell such as the MViBs using their CLEM approach. And I can accept that the images in Fig 3 and Vid 6-7 show the exiting of viral entities from the cell. But I don’t see how the authors’ can say that the material exiting the cell in the live images at this lower magnification is the same material they call MViBs in the higher magnification analyses. Especially since the heterogeneity of the MViBs structures seems to defy clear definition other than a vesicle containing a wide range of number and types of viral particles. Again maybe I am missing some aspect of the analyses.

• Line 333-334. I’m not sure I understand the practicality of the authors’ distinction between HCMV “using” MVBs by “transforming” them in to MViBs, versus “generated de novo.” Given the massive effect that HCMV infection seems to have on the secretory organelles ( a point well-appreciated by the authors; lines 77-87), this distinction seems to dissolve.

• Line 363-365. This statement implies that AD169 does not produce cell-associated virus, which depending on ones’ definition of “cell-associated”, is simply not true. One can certainly get copious amounts of infectious virus by sonication of AD169-infected cells. Likewise, we showed in Shultz JVI 2020 that TB40-BAC4-infected cells contain plenty of infectious virus, yet the spread of TB40-BAC4 was predominately via a cell-free mode; sensitive to neutralizing antibodies. We did not analyze AD169 in these studies, but I think the literature is replete with examples of AD169 spreading the presence of neutralizing antibodies, indicting some kind of cell-to-cell spread mode. Moreover, I’m not sure how the cited Murrell papers would support the authors’ statements. It is true that the BAC clone Merlin is especially good at cell-to-cell spread mechanisms, but I don’t think we understand why. Murrell 2017 suggests it is due to the high levels of gH/gL/UL128-131, but Shultz 2020 shows that is it not that simple. There could be a connection between the formation and trafficking of the MViB described by the authors’ and these spread mode differences.

• Line 411. As before, I would caution against making universal statements about HCMV biology from a study like Scriviano 2012, which involved only a single strain like TB40/e. in Schultz 2020, we demonstrated dramatic differences between the BAC clones TB40/e, TR and Merlin in cell-free and cell-associated virus in fibroblasts and epithelial cells. We did not investigate endothelial cells, but it seems likely that substantial inter-strain variation exists in this cell type also.

Reviewer #2: see above

Reviewer #3: Lines 101-203..."the deletion of the viral protein UL135 leads to an abrogation of virus-filled MVB-like structures, and mutating UL71, a viral protein likely being involved in membrane scission". No citation is provided to support the claim about UL135. Furthermore, UL133-138 are dispensable for replication in culture. So how strong is this claim in the first place? Aren't MVB-like structures seen with laboratory adapted strains like AD169 which lack UL135? Sounds like a strain-specific / cell type specific defect that should be contextualized as such and interpreted more carefully. In fact, defects in MVB formation secondary to UL135 could be indirect, since UL135-null viruses have defects in viral DNA synthesis and viral gene expression.

Line 288: "our data indicate a novel functional egress pathway for HCMV in which MViBs are targeted for secondary envelopment and subsequently exocytosed" .. this line is imprecise. Virions within MViBs ("multi-viral bodies") would be expected to already have undergone secondary envelopment.

Line 176-177: the idea that these large vesicles containing many virions might result from a so-far unrecognized "degradation pathway" seems far fetched and it's a little tiresome to see the argument given equal weight. Seems like a straw man.

PLOS authors have the option to publish the peer review history of their article (what does this mean?). If published, this will include your full peer review and any attached files.

Reviewer #1: **Yes: **Brent Ryckman

Reviewer #2: No

Reviewer #3: No
---

## [Editor Report · Decision Letter 1]

6 May 2022

Dear Prof. Dr. Bosse,

We are pleased to inform you that your manuscript 'Intermittent Bulk Release of Human Cytomegalovirus' has been provisionally accepted for publication in PLOS Pathogens.

Best regards,

Eain A Murphy, Ph.D.

Associate Editor

PLOS Pathogens

Shou-Jiang Gao

Section Editor

PLOS Pathogens

Kasturi Haldar

Editor-in-Chief

PLOS Pathogens

orcid.org/0000-0001-5065-158X

Michael Malim

Editor-in-Chief

PLOS Pathogens

orcid.org/0000-0002-7699-2064

Dr. Bosse,

Thank you for your resubmitted manuscript. After an editorial review, we feel this manuscript has been significantly improved and is now at a state that is acceptable to PLoS Pathogens.

Congratulations on a nice piece of work.

Cheers.

Eain Murphy
---

## [Editor Report · Acceptance letter]

20 Jul 2022

Dear Prof. Dr. Bosse,

We are delighted to inform you that your manuscript, "Intermittent Bulk Release of Human Cytomegalovirus," has been formally accepted for publication in PLOS Pathogens.

Best regards,

Kasturi Haldar

Editor-in-Chief

PLOS Pathogens

orcid.org/0000-0001-5065-158X

Michael Malim

Editor-in-Chief

PLOS Pathogens

orcid.org/0000-0002-7699-2064